# Optimal relationship between power and design driving loads for wind turbine rotors using 1D models

Kenneth Loenbaek[1,2], Christian Bak[2], Jens I. Madsen[1], and Bjarke Dam[1]

[1]Suzlon Blade Science Center, Havneparken 1, 7100 Vejle, Denmark
[2]Technical University of Denmark, Frederiksborgvej 399, 4000 Roskilde, Denmark

**Correspondence:** Kenneth Loenbaek (kenneth.loenbaek@suzlon.com)

**Abstract.** We investigate the optimal relationship between the aerodynamic power, thrust loading, and size of a wind turbine rotor when its design is constrained by a static aerodynamic load. Based on 1D-axial momentum theory, the captured power $\tilde{P}$ for a uniformly loaded rotor can be expressed in terms of the rotor radius $R$ and the rotor thrust coefficient $C_T$. Common types of static Design Driving Load Constraints (DDLC), e.g. limits on the permissible root-bending moment or tip deflection, may be generalized into a form that also depends on $C_T$ and $R$. The developed model is based on simple relations and make explorations of overall parameters possible in the early stage of the rotor design process. Using these relationships to maximize $\tilde{P}$ subject to a DDLC, shows that operating the rotor at the Betz limit (maximum $C_P$) does not lead to the highest power-capture. Rather, it is possible to improve performance with a larger rotor radius and lower $C_T$ without violating the DDLC. As an example, a rotor design driven by a tip-deflection constraint may achieve $1.9\%$ extra power-capture $\tilde{P}$ compared to the baseline (Betz limit) rotor.

The method is extended for optimization of rotors with respect to Annual Energy Production ($AEP$), where the thrust characteristics $C_T(V)$ needs to be determined together with $R$. This results in a much higher relative potential for improvements since the constraint limit can be met over a larger range of wind speeds. For example, a relative gain in $AEP$ of $+5.7\%$ is possible for a rotor design constrained by tip deflections compared with a rotor designed for optimal $C_P$. The optimal solution for $AEP$ leads to a thrust curve with three distinct operational regimes and so-called thrust-clipping.

**Keyword:** Wind Energy, Wind Turbine, Initial rotor design, Low Induction Rotor, Thrust-clipping, Peak-shaving

## 1 Introduction

From the inception of the wind energy industry, it has been a clear trend that rotor sizes are increasing. However, as discussed in Sieros et al. (2012), increasing the rotor size is not a clear way to decrease the Cost of Energy (CoE), since the rotor weight (closely related to rotor cost) will always scale with a higher exponent than the increase in power does. It is therefore argued that the lower CoE, that has taken place, is mostly due to technology improvements. The structural design of the turbine is built to carry the loads coming from the aerodynamics (steady or extreme) and the self-weight. Therefore lowering the loads should lead to a lighter blade. The steady aerodynamic load is applied to extract power and increasing the load leads to a higher power until the maximum power coefficient (max $C_P$) is reached. Increasing the load should lead to a heavier blade but it also leads to

higher power production. It goes to show that understanding the relationship between loading, power production and structural response is very important to get the most cost-effective turbine. It follows a trend that has been in the recent years that wind turbine optimization should include a more holistic approach with concepts like Multidisciplinary Design Analysis and Optimization (MDAO) and System Engineering (Bottasso et al., 2010; Zahle et al., 2015; Fleming et al., 2016; Perez-Moreno et al., 2016) here all the parts of the turbine design that affect the cost should be taken into account with the overall objective of minimizing CoE. Some of these related works focus more on how the rotor loading affects the power and the structural response. One of the concepts that come out of it is the so-called *Low-Induction-Rotor* (LIR) where the velocity induction at the rotor plane is lower than the value that maximizes the power coefficient. The concept was introduced by Chaviaropoulos and Sieros (2014) where it comes out of optimizing Annual Energy Production (AEP) by allowing the rotor to grow while constraining the flap root bending moment to be the same as a baseline. They state that the method can increase the AEP with 3.5% with a 10% increase in the rotor radius hereby showing that LIR can increase AEP while keeping the same flap root bending moment. It agrees with Kelley (2017) who allowed for a change in the radial loading resulting in an AEP increase of 5% with a radius increase of 11%. It was also investigated by Bottasso et al. (2015) where they both tested the potential of using LIR for AEP improvements with load constraint as well as a cost-optimized rotor. They find the same as the previous two investigations that LIR can improve AEP, but when they consider the CoE they find the LIR is not cost-effective, meaning that the additional cost of extending the blade is not compensated by the increase in power. This conclusion is opposed to the conclusion made by Buck and Garvey (2015b) where they target to minimize the ratio between Capital Expenditures (CapEx) and AEP. They arrive at LIR as the optimal solution for CapEx/AEP which is taken as a measure for CoE. Overall it seems that LIR can increase AEP while keeping the same load as a non-LIR baseline, but it is not clear if LIR is a cost-effective solution.

Another concept that is relevant in the context of this paper is Thrust-Clipping (also known as peak-shaving or force-capping). For turbines, it is often the case that the maximum thrust is reached just before rated power resulting in a so-called thrust peak. When using thrust-clipping this peak is lowered at the cost of power. It is used for many contemporary turbines for load alleviation, but is often added as a feature after the design process. Buck and Garvey (2015a) made a design study where they found that lowering the maximum thrust by 11% leads to 9% reduction in material content, at the cost of 0.1% lifetime energy, resulting in an overall reduction of 0.2% in cost of energy. Which shows that including thrust-clipping in the design process can lead to a lower CoE.

In this paper, we investigate the relationship between load, power and structural response of wind turbine rotors. Simple analytical models, based on 1D-aerodynamic-momentum theory and Euler-Bernoulli-Beam theory, are introduced to establish the first order relationship between these responses. This provides a useful framework for initial rotor design, e.g. when high level design parameters such as the rotor radius need to be fixed or to understand how load/structural responses will change with rotor size. The effect on the power curve and the related load/structural response with the variation in wind speeds is also investigated, which is useful for initial design of the highly coupled Aero-Servo-Elastic rotor design problem.

The relatively simple models used in this paper do not capture the full complexity needed for detailed wind turbine rotor design and should be considered a tool for early stage rotor design and overall exploration only. For example, the underlying theories (of 1D-aerodynamic-momentum and Euler-Bernoulli beams) assume steady-state conditions, while designs are often

constrained by load cases that are linked with extreme, unsteady, or non-normal operational events, e.g. extreme turbulence, gusts, emergency shutdowns, subsystem faults, or parked conditions. This is a limitation of the developed model, but if there is a relation between the steady-state loads and the extreme loads, which is very likely, then the results are still valid.

As mentioned before, the overall target for current turbine design is to lower the CoE, but a cost model is not used, which is also a limitation of this study. However, cost models relate to several assumptions made in the design process such as the price

of components in the design or composite lay-up of the blades, so a predicted cost will always be made with some uncertainty. Instead, load constraints are considered, much like for the above-mentioned Low-Induction-Rotor (LIR) example. As it was found by Bottasso et al. (2015), a constrained load might not lead to a lower CoE. So to accommodate for this, a constraint with fixed mass is made, which is thought to be a better approximation for a fixed cost.

This study is carried out to obtain an overview of how the rotor design more fundamentally is influenced by different types of

aerodynamic loading. Thus, an issue like the "self-weight" is important for modern turbines, but is not directly included in this study; especially the static-mass-moment has an impact on contemporary turbines. It could be included, but it was excluded to keep the study as simple as possible. Further discussion about the limitations and possible improvements of the study is given later in section 4.5.

## 2   Theory

This section will introduce the variables and the basic relationships used in this paper. It is split into two subsections: where subsection 2.1 introduces aerodynamic variables, equations, as well as the baseline rotor, while the second subsection 2.2 present scaling laws used to formulate design driving load constraints relative to the baseline rotor.

### 2.1   Aerodynamics

The theory for this Aerodynamics section is found in Sørensen (2016).

For wind turbine aerodynamics non-dimensional coefficients are often introduced and some of the common ones are for the rotor thrust ($C_T$) and power ($C_P$).

$$C_T = \frac{T}{\frac{1}{2}\rho V^2 \pi R^2} \tag{1}$$

$$C_P = \frac{P}{\frac{1}{2}\rho V^3 \pi R^2} \tag{2}$$

Where $T$ and $P$ are the rotor thrust and power respectively, $\rho$ is the air density, $V$ is the undisturbed flow speed and $R$ is the

rotor radius.

These definitions can be applied for any wind turbine rotor, but in this paper, we will use a simplified relationship between $C_T$ and $C_P$, which is derived from classical 1D-momentum theory. This implies an assumption of uniform aerodynamic loading across the rotor plane. The classical equations are often given in terms of the axial induction ($a$), which is defined as $a = 1 - \frac{V_{rotor}}{V}$ where $V_{rotor}$ is the axial flow speed in the rotor plane. By combining the two classical momentum theory

expressions for $C_P(a)$ and $C_T(a)$ (Sørensen, 2016, p. 11 eq. 3.8), the following relationship between these coefficients is arrived at:

$$C_T(a) = 4a(1-a) \atop C_P(a) = 4a(1-a)^2 \Bigg\} \implies C_P(C_T) = (1-a)C_T = \frac{1}{2}\left(1+\sqrt{1-C_T}\right)C_T, \quad C_T \in [0,1[ \tag{3}$$

Where $a(C_T)$ is found by inverting $C_T(a)$ and using the negative solution. A plot of $C_T$ vs. $C_P$ can be seen in figure 1. This

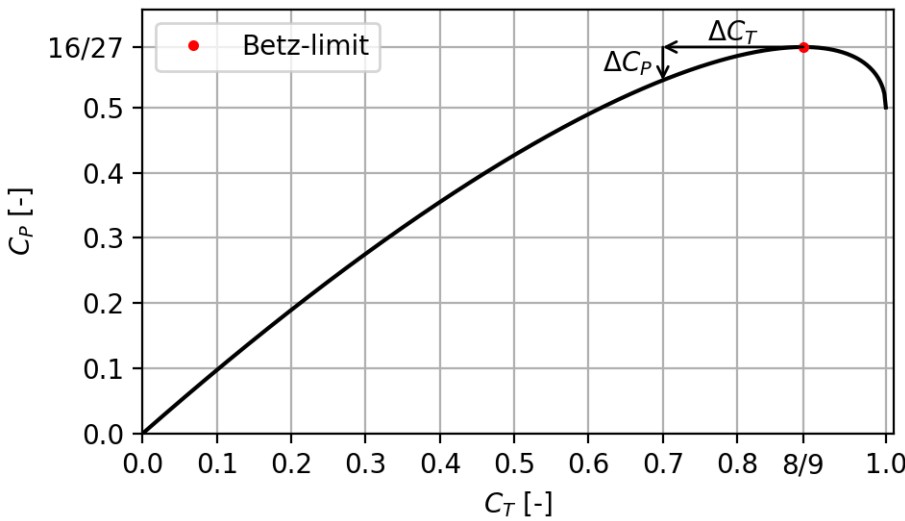

**Figure 1.** Relationship between normalized rotor load $C_T$ and power coefficient $C_P$ from one-dimensional momentum theory. Notes that around Betz-limit a small change in $C_T$ does not lead to a proportional change in $C_P$, this is illustrated by $\Delta C_T$ and $\Delta C_P$.

$C_P(C_T)$ curve is monotonically decreasing in slope and reaches a maximum $C_P = 16/27$ corresponding to the well-known
Betz-limit at $C_T = 8/9$. These monotonicity properties lead to the key observation that a reduction in thrust $(C_T = 8/9 - \Delta C_T)$ will not lead to a proportional change in power $(\Delta C_P)$. This motivates this paper's investigation of the trade-off between power and loads.

**Power-capture and Annual Energy Production (AEP)**

One way to understand the power yield of a rotor is to consider equation 2 as consisting of three separate terms:

$$100 \quad P = \underbrace{\frac{1}{2}\rho V^3}_{\text{Wind}} \cdot \underbrace{\pi R^2}_{\text{Size}} \cdot \underbrace{C_P}_{\text{Coefficient}} \tag{4}$$

*Wind* is the part of the equation that depends on the wind conditions, *Size* is the part of the equation that depends on the rotor swept area, and *Coefficient* is the part of the equation related to the power coefficient, representing the capability of the rotor to

extract power from the wind. The combination of equations 2 and 3 provides an expression that captures the latter two terms, which are the only ones affected by the design of the turbine:

$$\tilde{P}(C_T, \tilde{R}) = \frac{P}{\frac{1}{2}\rho V^3 \pi R_0^2} = C_P \tilde{R}^2 = \frac{1}{2}\left(1 + \sqrt{1 - C_T}\right) C_T \tilde{R}^2 \tag{5}$$

Where $\tilde{R}$ equals $R/R_0$, with $R_0$ being the radius of the baseline rotor. This equation will be referred to as the *Power-Capture* equation. It shows that power can be changed by changing either the loading ($C_T$) or the rotor radius ($R$). This will serve as the basic equation when the power-capture is optimized for a single design point.

When considering turbine design over the range of operational conditions, the *Annual Energy Production* (AEP) is introduced as an integral metric stating the energy produced per year given some wind speed frequency distribution. It can be computed as the power production ($P$) weighted by the probability density of wind speeds ($PDF_{wind}$) multiplied by the period of one year ($T_{year}$):

$$AEP = T_{year}\frac{1}{2}\rho\pi R_0^2 \int_{V_{CI}}^{V_{CO}} \tilde{P}(C_T(V), \tilde{R}) \cdot V^3 \cdot PDF_{wind}(V) dV \tag{6}$$

The wind speed probability distribution $PDF_{wind}$ will be described with a Weibull distribution. $V_{CI}$ and $V_{CO}$ is the wind speed for *Cut In* and *Cut Out* for wind turbine operation. Here they are taken to be $V_{CI} = 3\text{ms}^{-1}$ and $V_{CO} = 25\text{ms}^{-1}$, which is common numbers for modern wind turbines.

In this paper, we will use a dimensionless measure for AEP which is equivalent to the so-called capacity factor, defined as follows:

$$A\tilde{E}P(C_T, \tilde{R}) = \frac{AEP}{T_{year}P_{rated}} = \frac{AEP}{T_{year}\frac{1}{2}\rho\pi R_0^2 \frac{16}{27}V_0^3} = \frac{27}{16}\int_{\tilde{V}_{CI}}^{\tilde{V}_{CO}} \tilde{P}(C_T(\tilde{V}), \tilde{R}) \cdot \tilde{V}^3 \cdot PDF_{wind}(\tilde{V}) d\tilde{V} \tag{7}$$

$\tilde{V}$ is a normalized wind speed given as $V = \tilde{V}V_0$ where $V_0$ is the wind speed at which the turbine reach rated power. In all of this paper it is taken to be $V_0 = 10\text{ms}^{-1}$. It should further be noted that $PDF_{wind}dV$ is dimensionless and non-dimensionalizing the AEP it also follows that $PDF_{wind}d\tilde{V}$ is dimensionless. In all of this paper $A\tilde{E}P$ is computed by discretization of the integral and computing the integral with the trapezoidal rule given as $\int_{\tilde{V}_{CI}}^{\tilde{V}_{CO}} f(\tilde{V}; C_T, \tilde{R})d\tilde{V} \approx \sum_{i=1}^{N} \frac{f(\tilde{V}_{i+1};C_T,\tilde{R})+f(\tilde{V}_i;C_T,\tilde{R})}{2}\Delta\tilde{V}_i$ where the discretization ($N$) was found to become insignificant with $N = 200$.

**Baseline rotor**

The work here aims at demonstrating improved rotor performance compared to a baseline design. This baseline design is chosen to be a turbine operating at the Betz-limit below rated wind speed and keeping a constant power above rated.

$$C_{T,0} = \frac{8}{9} \approx 0.889, \qquad\qquad C_{P,0} = \frac{16}{27} \approx 0.593 \tag{8}$$

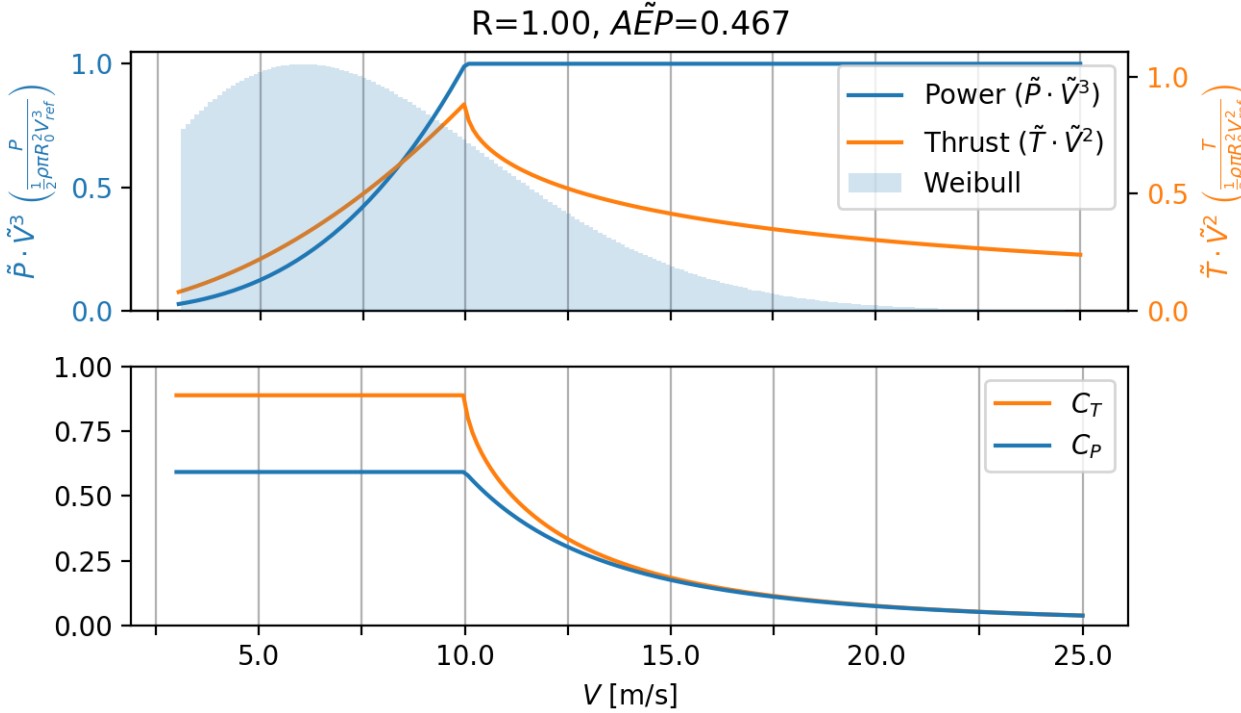

**Figure 2.** Top: The dimensionless power and thrust for the baseline rotor as a function of wind speed. Overlaid (in blue) the Weibull wind speed frequency distribution used throughout (IEC-class III: $V_{avg} = 7.5$, $k = 2$). Bottom: $C_T$ and $C_P$ as a function of wind speed. These curves reflect how most turbines are operated today, targeting maximum power coefficient below rated power, which leads to a thrust peak just before rated power.

This choice of baseline mimics the typical practice of designing wind turbines to target operation with maximum $C_P$ below rated power. In reality, turbines will not achieve maximum $C_P$ at $C_T = 8/9$ since losses alter the relationship between $C_T$ and $C_P$, but this does not change the fact that turbines are operated at the point of maximum $C_P$. Figure 2 shows the power and thrust curves for the baseline rotor.

In this paper, all results presented as the change in performance relative to that of the baseline rotor. For this reason, all the relevant variables will be normalized by the corresponding baseline rotor values.

$$\Delta R = \frac{R}{R_0} - 1 \tag{9}$$

$$\Delta \tilde{P} = \frac{C_P R^2}{C_{P,0} R_0^2} - 1 \tag{10}$$

$$\Delta \tilde{L} = \frac{C_T R^{L_{exp}}}{C_{T,0} R_0^{L_{exp}}} - 1 \tag{11}$$

$$\Delta A\tilde{E}P = \frac{A\tilde{E}P}{A\tilde{E}P_0} - 1 \tag{12}$$

where $\tilde{L}$ as well as $L_{exp}$ is a generalized load that is introduced in section 4.1 *(Effects on loads)* and it is written here for later reference.

## 2.2   Scale laws and constraints for Design Driving Loads

In this section, examples of static aerodynamic *Design Driving Loads* (DDL) will be presented. These examples are not meant to be exhaustive but include several of the key considerations that constrain the practical design of wind turbine rotors. From the scaled loads, *Design Driving Load Constraints* (DDLC) are introduced, which limit loads so that these do not exceed the levels of the baseline rotor. Based on the DDL examples, it is shown that DDLCs can be elegantly put in a generalized form.

**Thrust ($T$)**

Thrust typically does not limit the design of the rotor itself, but more likely is a constraint imposed from the design of the tower and/or foundation. The thrust scaling and the associated DDLC is given by:

Scaling                                                        DDLC

$$T = \frac{1}{2}\rho V_0^2 \pi R^2 C_T \qquad \Longrightarrow \qquad \text{DDLC}(T) = \frac{T}{T_0} = \frac{C_T}{C_{T,0}}\left(\frac{R}{R_0}\right)^2 \le 1 \tag{13}$$

**Root flap bending moment ($M_{flap}$)**

The root flap moment is the bending moment at the rotational center in the axial flow direction. To compute $M_{flap}$, the 1D-momentum-theory relations for infinitesimal thrust ($\mathrm{d}T$) and moment ($\mathrm{d}M$) are integrated:

$$\mathrm{d}T = \frac{1}{2}\rho V^2 C_T 2\pi r \mathrm{d}r \tag{14}$$

$$\mathrm{d}M_{flap} = r\mathrm{d}T \tag{15}$$

Where $r$ is the radius location of the infinitesimal load ($r \in [0, R]$). The moment scaling and DDLC can be found as:

Scaling

DDLC

$$M_{flap} = \int\limits_{0}^{R} \mathrm{d}M_{flap} = \frac{1}{3}\rho V_0^2 C_T \pi R^3 \qquad \Longrightarrow \qquad \mathrm{DDLC}(M_{flap}) = \frac{M_{flap}}{M_{flap,0}} = \frac{C_T}{C_{T,0}}\left(\frac{R}{R_0}\right)^3 \leq 1 \qquad (16)$$

As it is seen $M_{flap}$ scales with $R^3$ so it grows faster than the power, which grows as $R^2$. $M_{flap}$ is important for the blade design since the flap-wise aerodynamic loads need to be transferred via the blade structure to the root of the blade.

**Tip deflection ($\delta_{tip}$)**

Tip deflection is a common DDLC for contemporary utility-scale turbines, where tip clearance between tower and blade may become critical because of relatively long and slender blades. To get an idea for how tip-deflection scales with changes in
loading and rotor radius, Euler-Bernoulli Beam Theory (Bauchau and Craig, 2009, p. 189 eq. 5.40) is used. For the problem here it takes the form:

$$\frac{\mathrm{d}^2}{\mathrm{d}r^2}EI\frac{\mathrm{d}^2\delta}{\mathrm{d}r^2} = \frac{\mathrm{d}T}{\mathrm{d}r} = \frac{1}{2}\rho V^2 C_T 2\pi r \qquad (17)$$

Where $\delta$ is the deflection in the flap-wise direction of the blade at location $r$. $EI$ is the stiffness of the blade at location $r$. For modern turbines the stiffness decrease towards the tip of the blade. To get an estimate for the stiffness it is assumed that
stiffness follows the size of the chord ($EI \propto c$). The chord is given by the equation in (Sørensen, 2016, p. 68 eq. 5.26) with an approximation for the outer part of the blade it can be found that $c \propto R/r$ which means that $EI \propto R/r$. An approximate model for $EI$ can be made that have $EI \propto R/r$:

$$EI(r) = \frac{EI_r}{1 + \left(\frac{EI_r}{EI_t} - 1\right)\frac{r}{R}} \qquad (18)$$

Where $EI_r$ is the stiffness at the root and $EI_t$ is the stiffness at the tip of the blade. As mentioned above for wind turbines
$EI_r > EI_t$.

With the equation for $EI$ equation 17 can be solved by indefinite integration where the integration constants are determined from the following boundary conditions:

$$\underbrace{\delta(r=0) = 0, \quad \frac{\mathrm{d}\delta}{\mathrm{d}r}(r=0) = 0}_{\text{Clamped root}} \qquad\qquad \underbrace{\frac{\mathrm{d}^2\delta}{\mathrm{d}r^2}(r=R) = 0, \quad \frac{\mathrm{d}^3\delta}{\mathrm{d}r^3}(r=R) = 0}_{\text{Free tip}} \qquad (19)$$

The resulting displacement solution becomes:

$$\delta = \frac{11\pi}{120}\frac{V^2\rho}{EI_r}C_T R^5\left(\frac{2}{33}\left(\frac{EI_r}{EI_t} - 1\right)\tilde{r}^6 + \frac{1}{11}\tilde{r}^5 - \frac{5}{11}\left(\frac{EI_r}{EI_t} - 1\right)\tilde{r}^4 + \frac{10}{11}\left(\frac{2}{3}\frac{EI_r}{EI_t} - \frac{5}{3}\right)\tilde{r}^3 + \frac{20}{11}\tilde{r}^2\right) \qquad (20)$$

$$= \frac{11\pi}{120}\frac{V^2\rho}{EI_r}C_T R^5 \delta_{shape}\left(\tilde{r}, \frac{EI_r}{EI_t}\right) \qquad (21)$$

Where the normalized radius ($\tilde{r} \in [0,1]$) has been introduced so that $r = R \cdot \tilde{r}$. The maximum deflection occurs at the blade tip ($\tilde{r} = 1$), which leads to the following scaling relation and DDLC for tip deflection:

Scaling                                                                                         DDLC

$$\delta_{tip} = \frac{11\pi}{120} \frac{V^2 \rho}{EI_r} C_T R^5 \delta_{shape}\left(\tilde{r}=1, \frac{EI_r}{EI_t}\right) \qquad \Longrightarrow \qquad \text{DDLC}(\delta_{tip}) = \frac{\delta_{tip}}{\delta_{tip,0}} = \frac{C_T}{C_{T,0}}\left(\frac{R}{R_0}\right)^5 \leq 1 \qquad (22)$$

Where it has implicitly been assumed that any change in stiffness needs to follow:

$$\frac{EI_r}{EI_t} = \frac{EI_r}{EI_{r,0}}\left(\frac{EI_{r,0}}{EI_{t,0}} + \frac{26}{7}\right) - \frac{26}{7} \qquad (23)$$

With the simplest way to satisfy this relation being that $EI_r = EI_{r,0}$ which gives $\frac{EI_r}{EI_t} = \frac{EI_{r,0}}{EI_{t,0}}$.

**Tip deflection with constant mass**

The final example of a DDL is also based on tip deflection but includes a condition to maintain a constant mass of the load-carrying structure of the blade. To this end the stylized spar-cap layout depicted in figure 3 is assumed. This layout consists of two planks. The stiffness of a spar-cap structure with homogeneous Young's-modulus ($E$) can be found from the stiffness of a

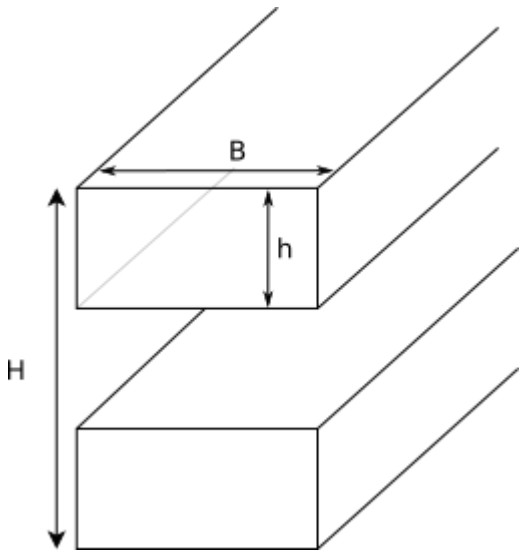

**Figure 3.** Assumed spar-cap structure with dimensions: $H$ is the total build height, $h$ is the space between planks, and $B$ is the plank width.

rectangle and the parallel axis theorem (see figure 3 for variable definition):

$$\left.\begin{array}{c} I_{rect} = \dfrac{Bh^3}{12} \\[2mm] EI = 2E\left(I_{rect} + A\left(\dfrac{H-h}{2}\right)^2\right) \\[2mm] A = Bh \end{array}\right\} EI = 2E\left(\frac{Bh^3}{12} + Bh\left(\frac{H-h}{2}\right)^2\right) = \frac{H^2 Bh}{2}\left(\frac{h^2}{3H^2} + \left(1 - \frac{h}{H}\right)^2\right) \qquad (24)$$

For modern wind turbines $h/H << 1$ meaning that a common approximation is:

$$EI \approx E\frac{H^2Bh}{2} \tag{25}$$

To compute the mass for such a structure it will be assumed that plank height $h$ and the plank width $B$ is constant and that the change in $EI$ comes from a decrease in building height $H$. If then $h$ is decreased when $R$ is increased the following relationship needs to be satisfied for the mass of the planks to have constant mass (assuming constant mass density):

$$Rh = R_0h_0 \tag{26}$$

From there it follows that changes in the radius of the rotor will change the stiffness as:

$$\left.\begin{array}{ll} EI \approx E\dfrac{H^2Bh}{2} & (25) \\[2mm] h = \dfrac{R_0h_0}{R} & (26) \end{array}\right\} \quad EI \approx E\frac{H^2BR_0h_0}{2R} \tag{27}$$

Combining the equation with the tip deflection equation (21) the following scaling and DDLC can be found:

$$\begin{array}{cc} \text{Scaling} & \text{DDLC} \end{array}$$

$$\left.\begin{array}{c} \delta_{tip} = \dfrac{11\pi}{120}\dfrac{V^2\rho}{EI_r}C_TR^5\delta_{shape}\left(\tilde{r}=1, \dfrac{EI_r}{EI_t}\right) \\[4mm] EI \approx E\dfrac{H^2BR_0h_0}{2R} \end{array}\right\} \implies \text{DDLC}(\delta_{tip+mass}) = \frac{C_T}{C_{T,0}}\frac{EI_{r,0}}{EI_r}\left(\frac{R}{R_0}\right)^5 = \frac{C_T}{C_{T,0}}\left(\frac{R}{R_0}\right)^6 \leq 1 \tag{28}$$

Where it has been used that changing $h$ by the same magnitude for the whole blade leads to $\frac{EI_r}{EI_t} = \frac{EI_{r,0}}{EI_{t,0}}$ and hereby not affecting $\delta_{shape}$. It should be noted that by choosing $B$ to change instead will lead to the same scaling, but with the difference being that changing the plank thickness might lead to higher-order effects, although they are expected to be insignificant.

**Generalizing the constraint form**

Considering the four DDLC examples presented above, there appears to be a pattern in the scaling relations that may be written as follows:

$$\frac{C_T}{C_{T,0}}\left(\frac{R}{R_0}\right)^{R_{exp}} \leq 1 \tag{29}$$

Where $R_{exp}$ is the *DDLC R-Exponent*.

If the constraint limit is met the following relationship can be written as:

$$R = R_0\left(\frac{C_{T,0}}{C_T}\right)^{\frac{1}{R_{exp}}} \tag{30}$$

## 3 Formulation of rotor design problems

Based on the performance and constraint relationships outlined in the previous section, this section will present the formulation for rotor design as optimization problems. Two different classes of problems are introduced, namely: *Power-Capture optimization* and *AEP optimization*, where the latter is a generalization of the former with the constraint depending on wind speed.

### 3.1 Power-Capture optimization

The optimization problem can be stated as:

$$\underset{C_T,\tilde{R}}{\text{maximize}} \quad \tilde{P} = \frac{1}{2}\left(1+\sqrt{1-C_T}\right)C_T\tilde{R}^2 \tag{31}$$

$$\text{subject to} \quad \frac{C_T}{C_{T,0}}\tilde{R}^{R_{exp}} \leq 1 \tag{32}$$

Where the definition of $\tilde{R} = R/R_0$ has been used for consistency . The solution for this optimization problem is presented in the 4.1 section.

It should be noted that this optimization problem is similar to the problem that is given by Chaviaropoulos and Sieros (2014) where they optimize while keeping $M_{flap}$. So the optimization problem in this paper is a generalization of their optimization problem.

### 3.2 AEP optimization

In contrast to the above mentioned optimization of power-capture, optimization with respect to AEP requires to determine $C_T(\tilde{V})$ so a function opposed to a scalar value. It is also necessary to fix the rated power to a constant value, while the wind speed at which rated power is reached is allowed to change. The problem can be formulated as:

$$\underset{C_T(\tilde{V}),\tilde{R}}{\text{maximize}} \quad A\tilde{E}P = \frac{27}{16}\int_{\tilde{V}_{CI}}^{\tilde{V}_{CO}} \tilde{P}(C_T(\tilde{V}),\tilde{R})\cdot\tilde{V}^3\cdot PDF_{wind}(\tilde{V})d\tilde{V} \tag{33}$$

$$\text{subject to} \quad \begin{aligned} &\tilde{V}^2\frac{C_T(\tilde{V})}{C_{T,0}}\tilde{R}^{R_{exp}} \leq 1, \quad \text{(DDLC)}\\ &\frac{27}{16}\tilde{P}(C_T(\tilde{V}),\tilde{R})\tilde{V}^3 \leq 1, \quad \text{(rated power)} \end{aligned} \tag{34}$$

Where the the wind speed scaling has been added to the DDLC.

## 4 Results and discussion

This section discusses the solutions to the rotor design optimization problems introduced in the previous section.

## 4.1 Optimizing for power-capture

The constrained optimization problem to maximize power-capture, as stated in section 3, may be simplified based on the observation that optimum solutions will occur at the DDL constraint limit. To understand this, consider that the power-capture of a rotor with an inactive constraint may always be improved by growing the rotor until the constraint is met. This is true irrespective of what DDLC that determines the rotor design. Hence, an explicit relation $\tilde{R}(C_T)$ can be used to reformulate from a constrained optimization problem in two variables to an unconstrained optimization problem in one variable.

$$\left.\begin{aligned} \tilde{P}(C_T, \tilde{R}) &= \frac{1}{2}\left(1 + \sqrt{1 - C_T}\right)C_T \tilde{R}^2 \quad (5) \\ \tilde{R} &= \left(\frac{C_{T,0}}{C_T}\right)^{\frac{1}{R_{exp}}} \quad (30) \end{aligned}\right\} \implies \tilde{P}(C_T) = \frac{C_{T,0}^{2\frac{1}{R_{exp}}}}{2}\left(1 + \sqrt{1 - C_T}\right)C_T^{1 - 2\frac{1}{R_{exp}}} \tag{35}$$

With the optimization problem now being:

$$\underset{C_T}{\text{maximize}} \quad \tilde{P} = \frac{C_{T,0}^{2\frac{1}{R_{exp}}}}{2}\left(1 + \sqrt{1 - C_T}\right)C_T^{1 - 2\frac{1}{R_{exp}}} \tag{36}$$

By differentiating the objective function 35 with respect to $C_T$ and finding its root, the optimal $C_T$ as a function of $R_{exp}$ is arrived at:

$$\frac{d\tilde{P}(C_T)}{dC_T} = 0 \implies \tag{37}$$

$$C_T = \frac{8\left(R_{exp}^2 - 3R_{exp} + 2\right)}{\left(3R_{exp} - 4\right)^2} \tag{38}$$

This unique solution is a maximum, which is apparent from the always positive signs of $\Delta P$ in figure 4. This figure shows the optimal solution for $C_T$ and $C_P$, as well as the relative change in radius ($\Delta R$) and power ($\Delta P$) compared to the baseline rotor. From the two left plots, $C_P$ is observed to approach the dashed baseline performance (Betz rotor) much faster than $C_T$ as $R_{exp}$ increases. This is a consequence of the relationship between $C_T$ and $C_P$ (figure 1). Especially around the Betz-limit, the gradient is very small, which means that changes in $C_T$ do not lead to proportional changes in $C_P$. Turning to the two plots on the right in figure 4, it is seen that the lower $C_P$ is more than compensated by increasing $R$ since the relative change in power ($\Delta P$) is always positive.

When maximizing power-capture for a given thrust ($R_{exp} = 2$; blue dashed vertical line in figure 4), it is found that $C_T \to 0$ and $\Delta R \to \infty$ while $\Delta P \to 50\%$, which was found by investigating the limit value behavior when $R_{exp} \to 2$. Since $\Delta R \to \infty$ is not of much practical interest, further explanation is not given here. Alternatively, the maximum power for a given flap root moment ($R_{exp} = 3$; orange line) may be achieved by increasing the rotor radius by $11.6\%$ compared to the baseline design (maximum $C_P$). The corresponding relative increase in power $\Delta P$ is $7.6\%$. Finally, designs constrained by tip-deflection ($R_{exp} = 5$; green line) allows the relative power $\Delta P$ to increase by $1.90\%$ with a relative change in radius $\Delta R$ of $2.30\%$. A table with the results for the the increase in power-capture ($\Delta P$) and radius ($\Delta R$) for 4 designs ($R_{exp} = 2, 3, 5, 6$) can be seen in figure 6. In conclusion, rotors with an active static aerodynamic DDLC should not be designed for maximum $C_P$ as more power can be generated by rotors with lower $C_T$ and a larger radius $R$, without violating the relevant DDLC.

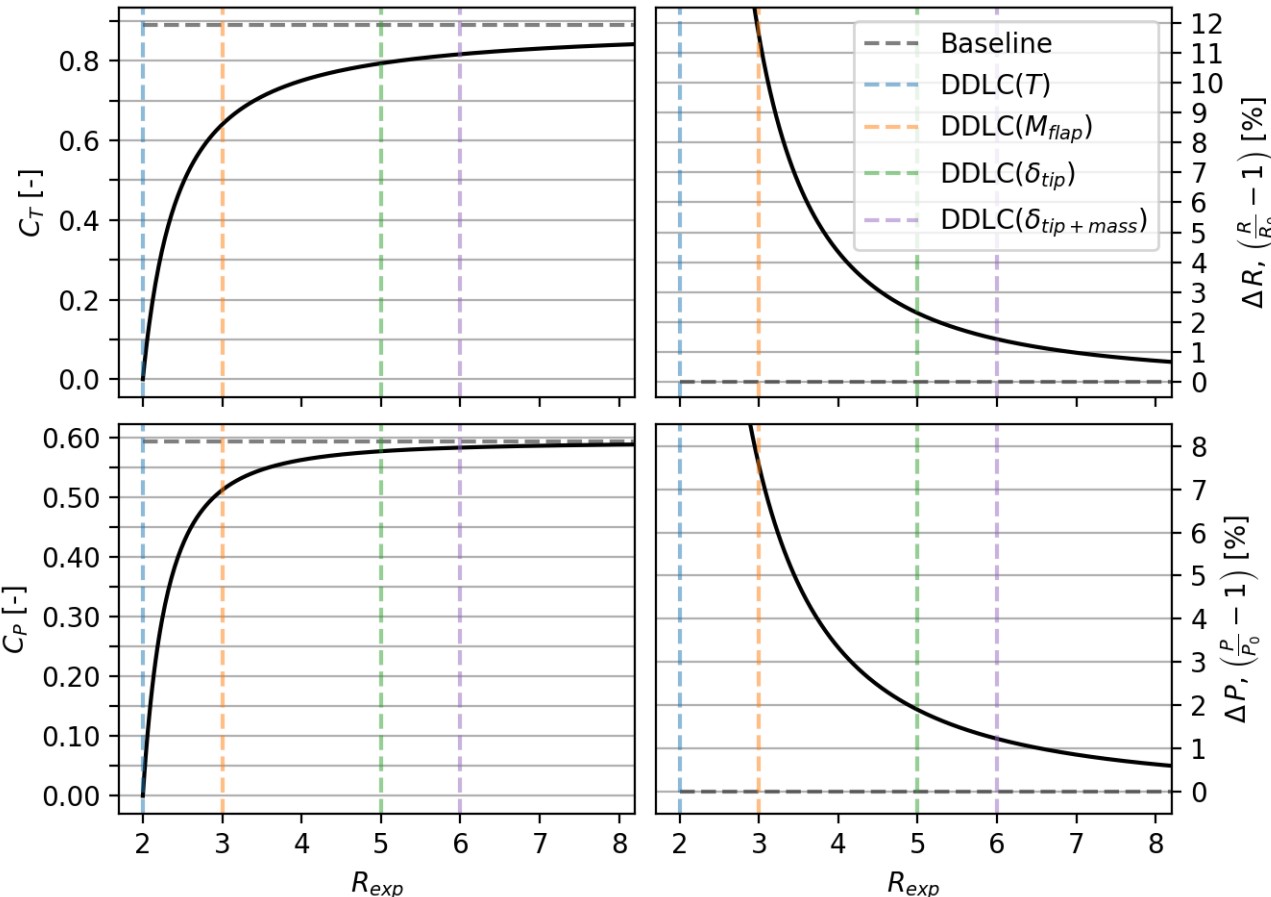

**Figure 4.** Top left: Optimal $C_T$ as a function of the constraint $R$-exponent ($R_{exp}$). Low left: $R_{exp}$ vs. $C_P$, notice that the optimal $C_P$ curve has a steeper slope and hugs the baseline closer than $C_T$. Top right: $R_{exp}$ vs. relative change in radius $\Delta R$. Lower right: $R_{exp}$ vs. relative change in power-capture ($\Delta \tilde{P}$). Despite the similar shape of curves a difference between the two is that $\Delta P(R_{exp} \to 2) = 50\%$ where $\Delta R(R_{exp} \to 2) \to \infty$. The vertical lines represent each of the example constraints. (*DDLC=Design Driving Load Constraint*).

**Effect on loads**

Even though meeting the constraint limits means that the chosen DDL will be the same as the baseline, it is interesting to know what happens to the loads that scale differently than the DDL. As an example, if the DDLC is $M_{flap}$ ($R_{exp} = 3$) it is given that it will not change relative to the baseline, but it could be interesting to know what happens to the $T$ and $\delta_{tip}$.

To investigate it we will introduce a *Generalized Load* ($L$) as a measure of how a load scale.

$$L = K_0 V_0^2 C_T R^{L_{exp}} \tag{39}$$

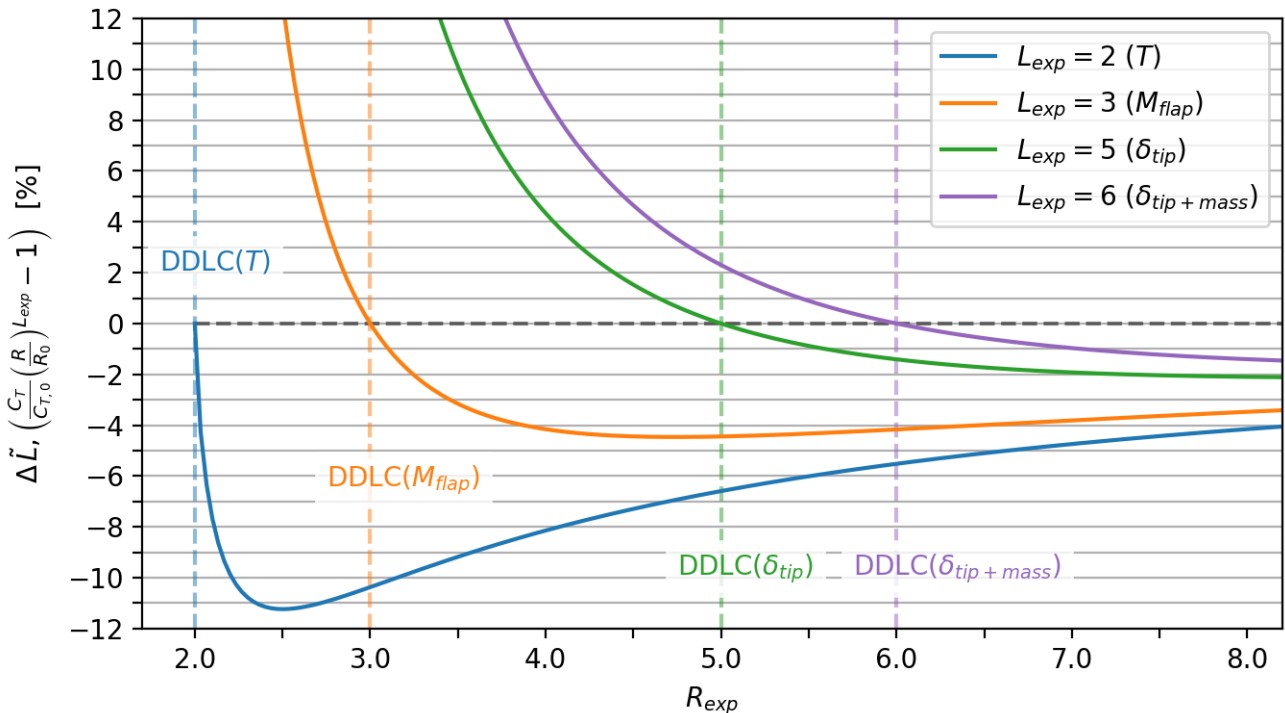

**Figure 5.** Relative change in different rotor load parameters ($\Delta\tilde{L}$) depending on DDLC. The scaling of loads have the form $\tilde{L} = C_T R^{L_{exp}}$, e.g. $L_{exp} = 2$ scales as the rotor thrust $T$ and $L_{exp} = 5$ scales as the tip deflection $\delta_{tip}$. Each curve depicts how a load parameter would change depending on design driving constraint. As an example consider a design limited by tip deflection DDLC($\delta_{tip}$), i.e. $R_{exp} = 5$ matching the dashed green line. Tip deflection meets requirements, while thrust ($T$) is lowered 6.6% and flap moment $M_{flap}$ by 4.4%.

Where $K_0$ is a scaling constant and $L_{exp}$ is the *Generalized Load Exponent*. The Generalized Load equation can be made
non-dimensional as:

$$\tilde{L} = \frac{L}{K_0 V_0^2 R_0^{L_{exp}}} = C_T \tilde{R}^{L_{exp}} \tag{40}$$

The difference between $L_{exp}$ and $R_{exp}$ is that $R_{exp}$ results in a design, wheres $L_{exp}$ is a load for a design. As an example take a design made for tip-deflection ($R_{exp} = 5$) then $L_{exp} = 3$ will describe the $M_{flap}$ load for that design.

An equation for the relative change $\Delta\tilde{L}$ can be found in terms of the baseline rotor as:

$$\left.\begin{array}{l} \tilde{L} = C_T \tilde{R}^{L_{exp}} \quad (40) \\[2mm] \tilde{R} = \left(\dfrac{C_{T,0}}{C_T}\right)^{\frac{1}{R_{exp}}} \quad (30) \\[2mm] \tilde{L}_0 = C_{T,0}\tilde{R}_0^{L_{exp}} = C_{T,0} \end{array}\right\} \implies \Delta\tilde{L} = \frac{\tilde{L}}{\tilde{L}_0} - 1 = \left(\frac{C_T}{C_{T,0}}\right)^{1-\frac{L_{exp}}{R_{exp}}} - 1 \tag{41}$$

Since it is known that $C_T \leq C_{T,0}$ the following can be concluded:

$L_{exp} < R_{exp}$                                         The load is lower than the baseline level

$L_{exp} = R_{exp}$                                         The load is identical to the baseline level

$L_{exp} > R_{exp}$                                         The load is larger than the baseline level

This agrees with figure 5, which illustrates the effect of design constraints (DDLC) on different loads. For example, consider tip-deflection ($R_{exp} = 5$, DDLC($\delta_{tip}$), dashed green line). Looking at the green solid line ($L_{exp} = 5$) it is seen that the relative change in $L$ is zero as expected. Now looking at the loads with $L_{exp} < R_{exp}$, namely thrust ($L_{exp} = 2$) and flap-moment ($L_{exp} = 3$) it is seen that $\Delta L$ is lower than the baseline with $\Delta T = -6.6\%$ and $\Delta M_{flap} = -4.4\%$. But the loads where $L_{exp} > R_{exp}$ the loads are increased. If there was a load that scaled like $L_{exp} = 6$ the load would be increased by $\Delta L_{(L_{exp}=6)} =$

$+2.3\%$. Furthermore, figure 5 shows that the relative decrease in load is always most pronounced for the thrust ($L_{exp} = 2$), the biggest impact occurring around $R_{exp} \approx 2.5$. All the relative change curves have distinct minima, but at the same time are characterized by large plateaus of relatively small change. Another observation is how quickly the curves grows for $L_{exp} > R_{exp}$. As an example take DDLC($M_{flap}$) in this case $\Delta \delta_{tip} = +24.5\%$ and $\Delta L_{(L_{exp}=6)} = +38.9\%$. The relative change in loads becomes smaller as $R_{exp}$ increases. A sketch with a zoomed view of the tip and a table with the values can be seen in

figure 6.

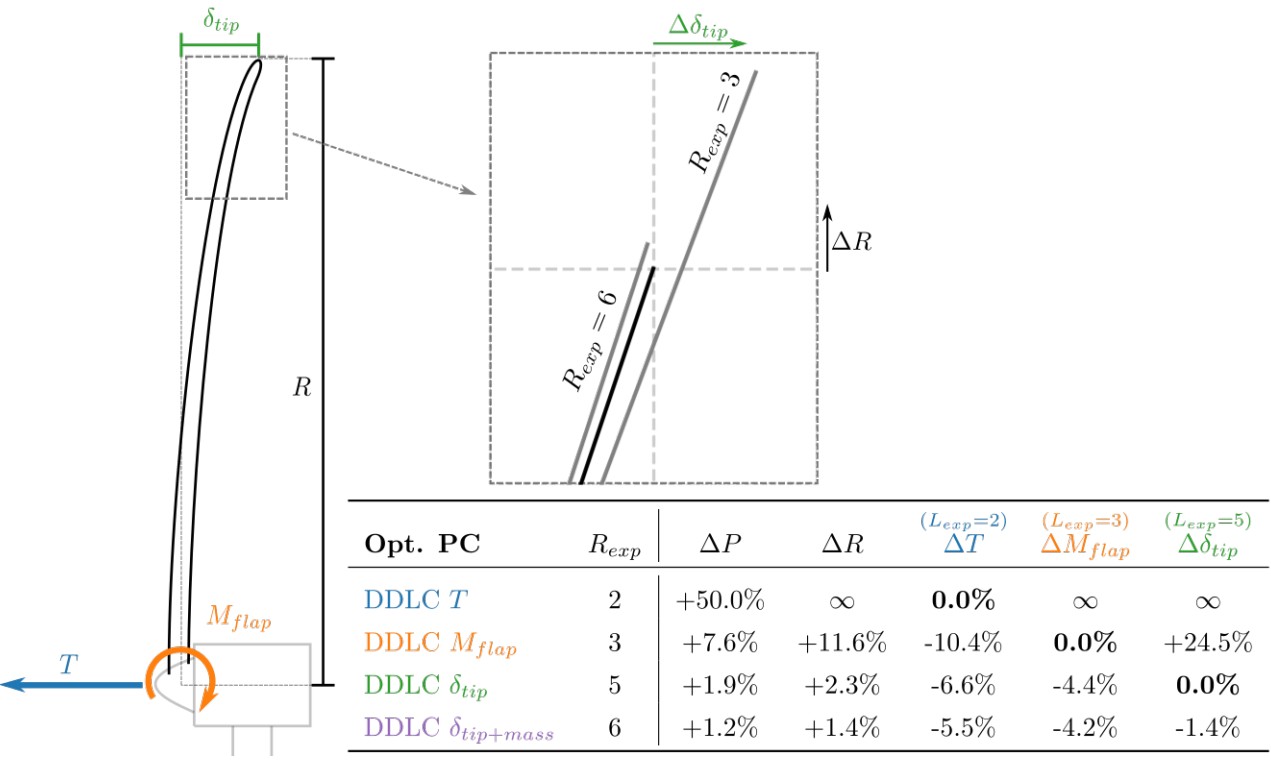

| Opt. PC | $R_{exp}$ | $\Delta P$ | $\Delta R$ | $(L_{exp}=2)$ $\Delta T$ | $(L_{exp}=3)$ $\Delta M_{flap}$ | $(L_{exp}=5)$ $\Delta \delta_{tip}$ |
|---------|-----------|-----------|-----------|--------------------------|---------------------------------|-------------------------------------|
| DDLC $T$ | 2 | +50.0% | $\infty$ | **0.0%** | $\infty$ | $\infty$ |
| DDLC $M_{flap}$ | 3 | +7.6% | +11.6% | -10.4% | **0.0%** | +24.5% |
| DDLC $\delta_{tip}$ | 5 | +1.9% | +2.3% | -6.6% | -4.4% | **0.0%** |
| DDLC $\delta_{tip+mass}$ | 6 | +1.2% | +1.4% | -5.5% | -4.2% | -1.4% |

**Figure 6.** Sketch of a turbine with the load/structural response outlined. The zoomed figure shows the radius increase ($\Delta R$) and the change in tip-deflection ($\Delta\delta_{tip}$) for two different DDLCs (bold black line is the baseline). The table shows the relative change in power, radius and load/structural response for different DDLCs. $R_{exp}=2$ is a thrust constraint design, $R_{exp}=3$ is a flap moment constraint design, $R_{exp}=5$ is a tip-deflection constraint design and $R_{exp}=6$ is tip-deflection+constant mass constraint design.

## 4.2 Low-Induction-Rotor

The concept was mentioned in the introduction since it has had some attention over the recent years. The Low-Induction-Rotors (LIR) are rotors designed with lower axial induction $a$ than the level that maximizes $C_P$. The concept is to a certain degree analogous with optimization of rotors for power-capture.

To investigate such an LIR design it is chosen to fix the $C_T$ value below rated power to be the same as for the power-capture optimization for a given $R_{exp}$. If the radius was set to the same value as for power-capture it will result in the constraint limit not being met since the turbine reaches rated power earlier. Since $C_T$ is fixed and the constraint limit needs to be met, then the wind speed at which the turbine reaches rated power ($\tilde{V}_{rated}$) can be found. It is found through the normalized power (the integrant of equation 7 without the $PDF_{wind}$) and the constraint limit with wind speed scaling (equation 30 multiplied with $\tilde{V}^2$):

$$\left. \begin{array}{c} \frac{27}{16}\frac{1}{2}\left(1+\sqrt{1-C_T}\right)C_T\tilde{R}^2\tilde{V}^3 = 1 \\ \tilde{V}^2 \frac{C_T}{C_{T,0}}\tilde{R}^{R_{exp}} = 1 \end{array} \right\} \implies \tilde{V}_{rated} = \left( \frac{16}{27}\frac{2}{\left(1+\sqrt{1-C_T}\right)C_T}\left(\frac{C_T}{C_{T,0}}\right)^{\frac{2}{R_{exp}}} \right)^{\frac{1}{3-\frac{4}{R_{exp}}}} \tag{42}$$

With the rated wind speed the rotor radius can be found using the following steps:

1)  $C_T = \frac{8\left(R_{exp}^2 - 3R_{exp}+2\right)}{(3R_{exp}-4)^2}$ $\tag{38}$

2)  $\tilde{V}_{rated} = \left( \frac{16}{27}\frac{2}{\left(1+\sqrt{1-C_T}\right)C_T}\left(\frac{C_T}{C_{T,0}}\right)^{\frac{2}{R_{exp}}} \right)^{\frac{1}{3-\frac{4}{R_{exp}}}}$ $\tag{42}$

3)  $\tilde{R} = \left( \frac{1}{\tilde{V}_{rated}^2}\frac{C_{T,0}}{C_T} \right)^{\frac{1}{R_{exp}}}$ $\tag{43}$

With $C_T$, $\tilde{V}_{rated}$ and $\tilde{R}$, $A\tilde{E}P$ can be computed using equation 7.

The LIR is illustrated by the examples in figure 7 and 8 where the present analysis framework has been applied with constraints pertaining to respectively flap moments ($R_{exp} = 3$) and tip deflections ($R_{exp} = 5$).

In both cases, the resulting power curves are slightly above the equivalent baseline ones, and the thrust peaks are reduced compared with the baseline. The relative change in $AEP$ results in a smaller change than the change in power at the design point. For the case with DDLC($M_{flap}$), $\Delta AEP = 6.0\%$ while the power-capture increased by $\Delta P = 7.6\%$. The corresponding improvements for a tip deflection constrained rotor (DDLC($\delta_{tip}$)) are $\Delta AEP = 1.2\%$ and $\Delta P = 1.9\%$. The lower relative improvement for the LIR is related to the amount of the power that is produced below rated power. The results for LIR are summarized in figure 9 with a table and a sketch showing the relative changes in $AEP$, radius, thrust, root-flap-moment and tip-deflection for 4 different designs ($R_{exp} = 2, 3, 5, 6$). From figure 9 the thrust constraint design (DDLC. $T$; $R_{exp} = 2$) is seen to have diverging values for $\Delta R$, $\Delta M_{flap}$ and $\Delta \delta_{tip}$. As it was the case for power-capture optimization these results are found from investigating the result in the limit where $R_{exp} \to 2$. Even though this result of $\Delta R \to \infty$ is interesting, the corresponding consequence of $\Delta M_{flap} \to \infty$ makes this infeasible for practical use, so this will not be studied further here.

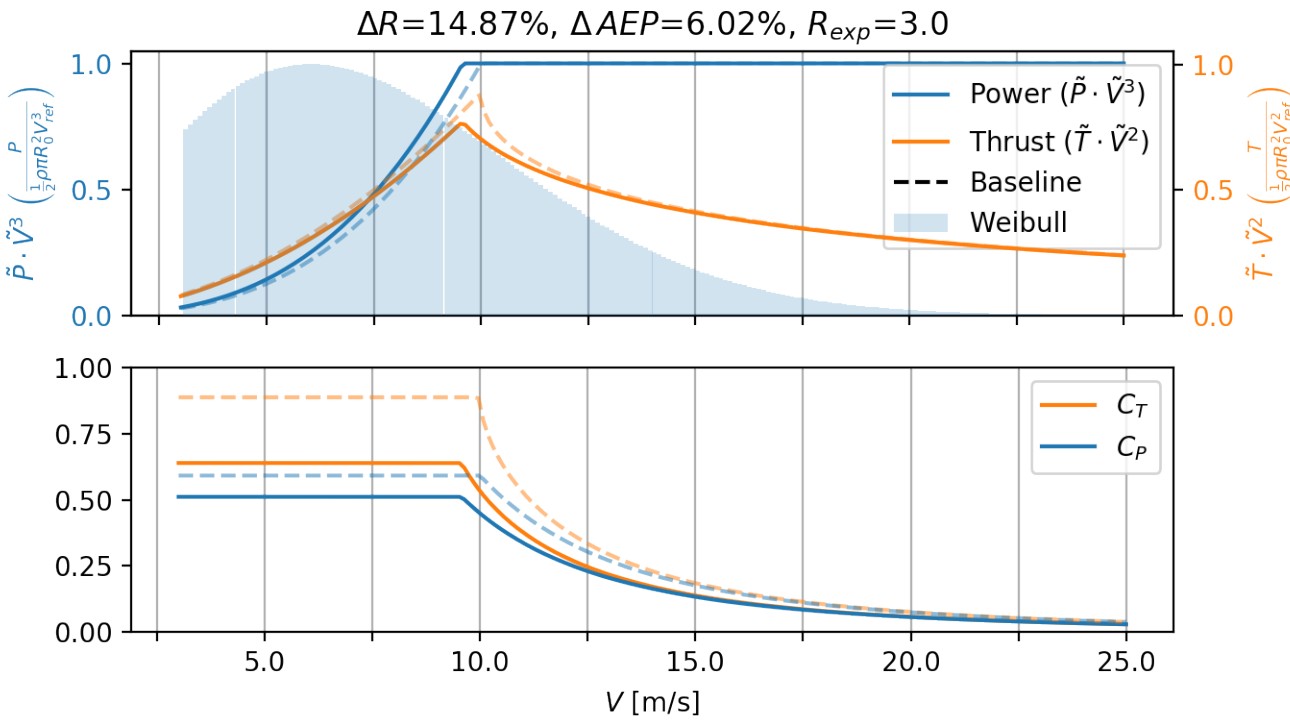

**Figure 7.** Power and thrust curves for Low-Induction-Rotor (solid lines), designed using the present method with DDLC exponent $R_{exp} = 3$, which corresponds to a $M_{flap}$ constraint. The dashed line is the baseline rotor optimized for max $C_P$.

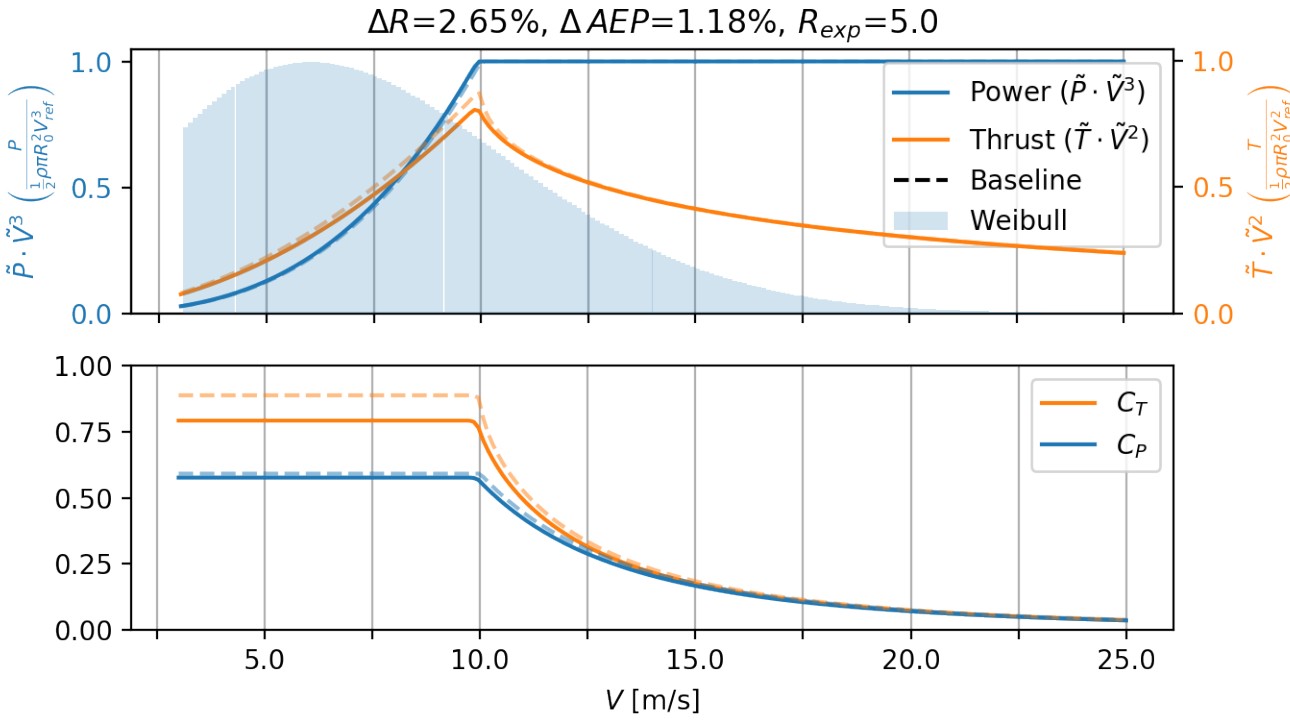

**Figure 8.** Power and thrust curves for rotor with DDLC exponent $R_{exp} = 5$ (solid lines), corresponding to a $\delta_{tip}$ constraint. The dashed line is the baseline rotor optimized for max $C_P$.

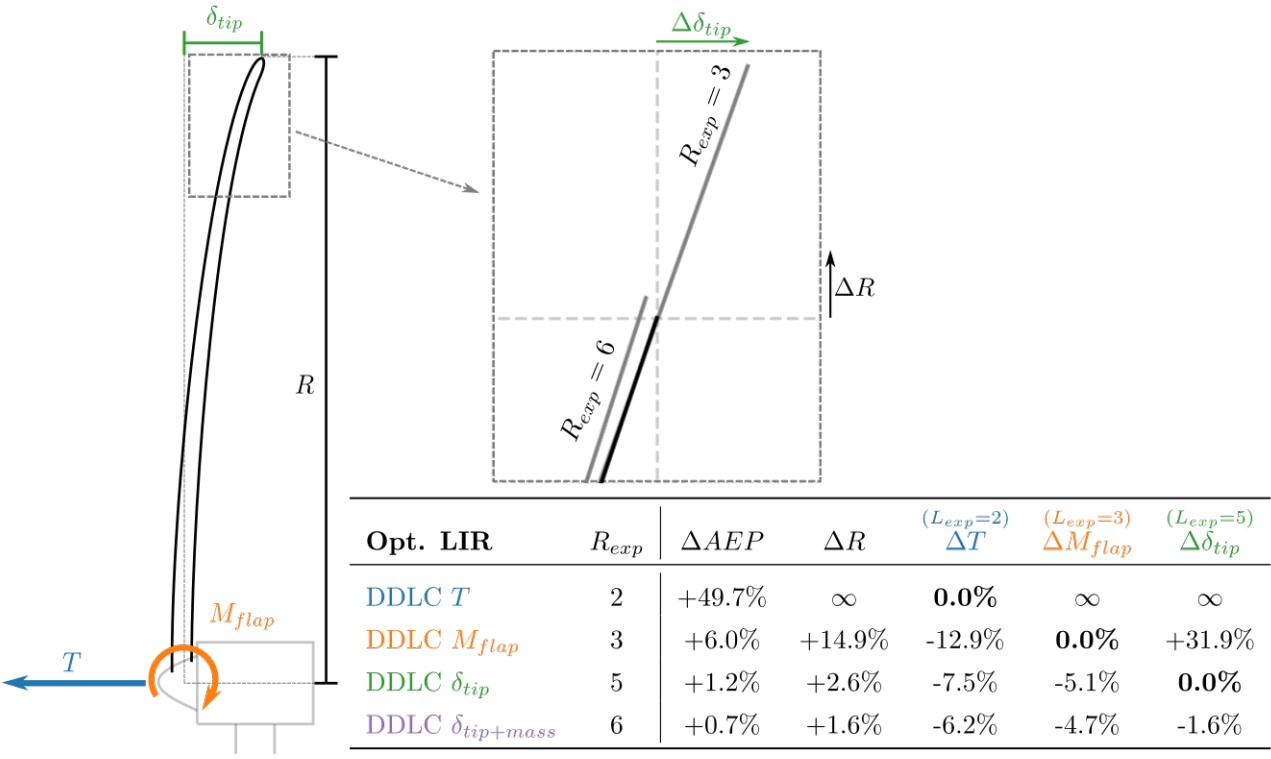

| Opt. LIR | $R_{exp}$ | $\Delta AEP$ | $\Delta R$ | $(L_{exp}=2)$ $\Delta T$ | $(L_{exp}=3)$ $\Delta M_{flap}$ | $(L_{exp}=5)$ $\Delta \delta_{tip}$ |
|---|---|---|---|---|---|---|
| DDLC $T$ | 2 | +49.7% | $\infty$ | **0.0%** | $\infty$ | $\infty$ |
| DDLC $M_{flap}$ | 3 | +6.0% | +14.9% | -12.9% | **0.0%** | +31.9% |
| DDLC $\delta_{tip}$ | 5 | +1.2% | +2.6% | -7.5% | -5.1% | **0.0%** |
| DDLC $\delta_{tip+mass}$ | 6 | +0.7% | +1.6% | -6.2% | -4.7% | -1.6% |

**Figure 9.** Sketch of a turbine with the load/structural response outlined. The zoomed figure shows the radius increase ($\Delta R$) and the change in tip-deflection ($\Delta \delta_{tip}$) for two different DDLCs (bold black line is the baseline). The table shows the relative change in power, radius and load/structural response for different DDLCs. $R_{exp} = 2$ is a thrust constraint design, $R_{exp} = 3$ is a flap moment constraint design, $R_{exp} = 5$ is a tip-deflection constraint design and $R_{exp} = 6$ is tip-deflection+constant mass constraint design.

## 4.3 AEP optimized rotor

As mentioned in section 3, the variables considered for optimization of $AEP$ are $C_T(\tilde{V})$ as well as $\tilde{R}$. In this formulation, $C_T$ can be adjusted independently for each wind speed, which ideally can be achieved through blade pitch control. The relative radius $\tilde{R}$ couples the rotor operation across all wind speeds, as it necessarily is constant. Based on initial studies, the optimizer targets solutions with three distinct operational ranges, which ordered by wind speed are:

- Operation with maximum power coefficient ($\max C_P$)

- Operation at constraint limit (constant thrust $T$)

- Operation at rated power

this can be used to make $C_T$ a function of $\tilde{R}$ hereby decreasing the optimization problem to an unconstrained optimization in one variable ($\tilde{R}$). The $C_T$ function is given as:

$$C_T(\tilde{V}, \tilde{R}) = \begin{cases} \frac{8}{9} & \frac{8}{9} \leq \tilde{V}^{-2} C_{T,0} \tilde{R}^{-R_{exp}} & (\max C_P) \\ \tilde{V}^{-2} C_{T,0} \tilde{R}^{-R_{exp}} & 1 \leq \frac{27}{16} \frac{1}{2} \left(1 + \sqrt{1 - C_T}\right) C_T \tilde{R}^2 \tilde{V}^3 & (\text{constraint limit}) \\ 1 = \frac{27}{16} \frac{1}{2} \left(1 + \sqrt{1 - C_T}\right) C_T \tilde{R}^2 \tilde{V}^3 & 1 > \frac{27}{16} \frac{1}{2} \left(1 + \sqrt{1 - C_T}\right) C_T \tilde{R}^2 \tilde{V}^3 & (\text{rated power}) \end{cases} \tag{44}$$

Where the last equation needs to be solved to get $C_T$, the solution is a third-order polynomial, which is easier solved numerically.

The only free parameter that needs to be determined to find the optimal $AEP$ is $\tilde{R}$. The optimization problem can be reformulated as:

$$\underset{\tilde{R}}{\text{maximize}} \quad A\tilde{E}P = \int_{\tilde{V}_{CI}}^{\tilde{V}_{CO}} \tilde{P}(C_T(\tilde{V}, \tilde{R}), \tilde{R}) \cdot \tilde{V}^3 \cdot PDF_{wind}(\tilde{V}) d\tilde{V} \tag{45}$$

The problem can be solved with most optimization solvers since the $AEP$ can be computed explicitly if $\tilde{R}$ is given. The optimization problem was solved with the L-BFGS-B algorithm described in Zhu et al. (1997) though the use of Scipy (Millman and Aivazis (2011)).

Examples of the resultant power and thrust curves can be seen in figure 10 and 11, for DDLC($M_{flap}$) and DDLC($\delta_{tip}$) respectively. Looking at figure 10 ($R_{exp} = 3$) it is clear that the power and thrust curves have changed quite substantially, compared to the baseline Betz-rotor (dashed curves). The thrust curve does not have a sharp peak anymore, but a flat plateau. As mentioned in the introduction this is often referred to as thrust-clipping. It comes from the DDLC equation 44 which shows that $C_T \propto \tilde{V}^{-2}$, and since thrust is proportional to $T \propto C_T \tilde{V}^2$ it means that the thrust is constant. As mentioned, the region where the rotor is thrust-clipped is also where the DDLC is active, so opposed to the baseline and LIR rotor the DDLC is active over a larger range of $V$. The larger range of $V$ is also part of why $\Delta R = 44.6\%$ which is a huge increase. As a result, it also leads to a large increase in $\Delta AEP = 19.9\%$. This is a very large change in $\tilde{R}$ and the feasibility of such a design is doubtful. As it is shown later the change in maximum loads (see figure 13) shows a significant change in loads with $L_{exp} > R_{exp}$.

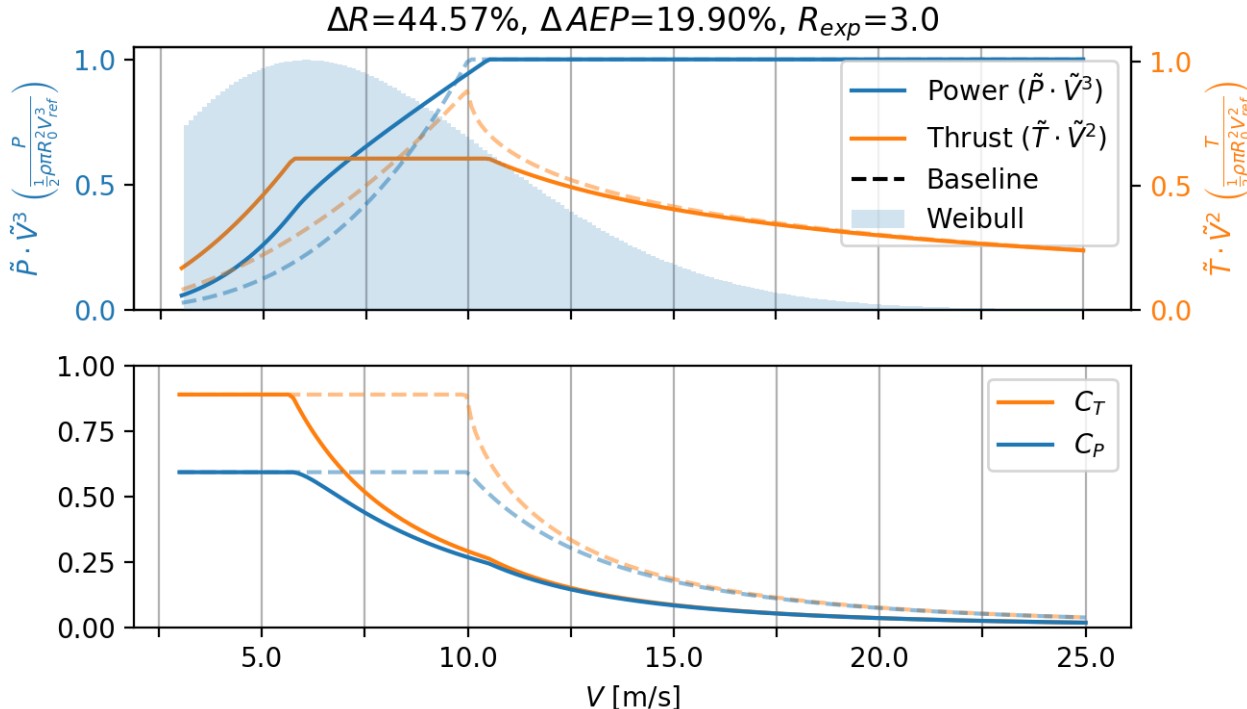

**Figure 10.** Power and thrust curve for AEP optimized rotor (solid lines) with DDLC exponent is $R_{exp} = 3$ which is equivalent to a constraint on $M_{flap}$. The dashed line is the baseline rotor optimized for max $C_P$ below rated power.

A more realistic design for modern turbines is found in figure 11 ($R_{exp} = 5$). Here the changes are less but still significant with $\Delta R = 10.7\%$ and $\Delta AEP = 5.8\%$. It shows the same shape with the thrust-clipped curve, but now it is over a smaller range of $V$. As mentioned in the introduction thrust-clipping is also found by Buck and Garvey (2015a) to be a beneficial way to lower CoE.

In figure 12 the relative change in $R$ and $AEP$ can be see as a function of the DDLC $R$-exponent. The plot both contains the result for the $AEP$-optimized rotor (AEP opt.; solid black line) and the Low-Induction-Rotor (LIR opt.; dash-dotted gray line). The difference between the two is significant especially for $\Delta AEP$. The results for the $AEP$ optimized rotor are summarized in figure 14 with a table and a sketch, that shows the relative changes. As it was the case for power-capture optimization and LIR optimization some values diverge when $R_{exp} \rightarrow 2$ and the results are found by investigating this limit. But since it has no practical value, further explanation is omitted here.

**Effect on loads**

In figure 13 a plot of the relative change in maximum loads as a function of the DDLC $R$-exponent. The relative max load ($\Delta \tilde{L}_{\max}$) is not comparing the loads at each $\tilde{V}$ but the max load for the baseline at $\tilde{V} = 1$ (rated wind speed) to the max

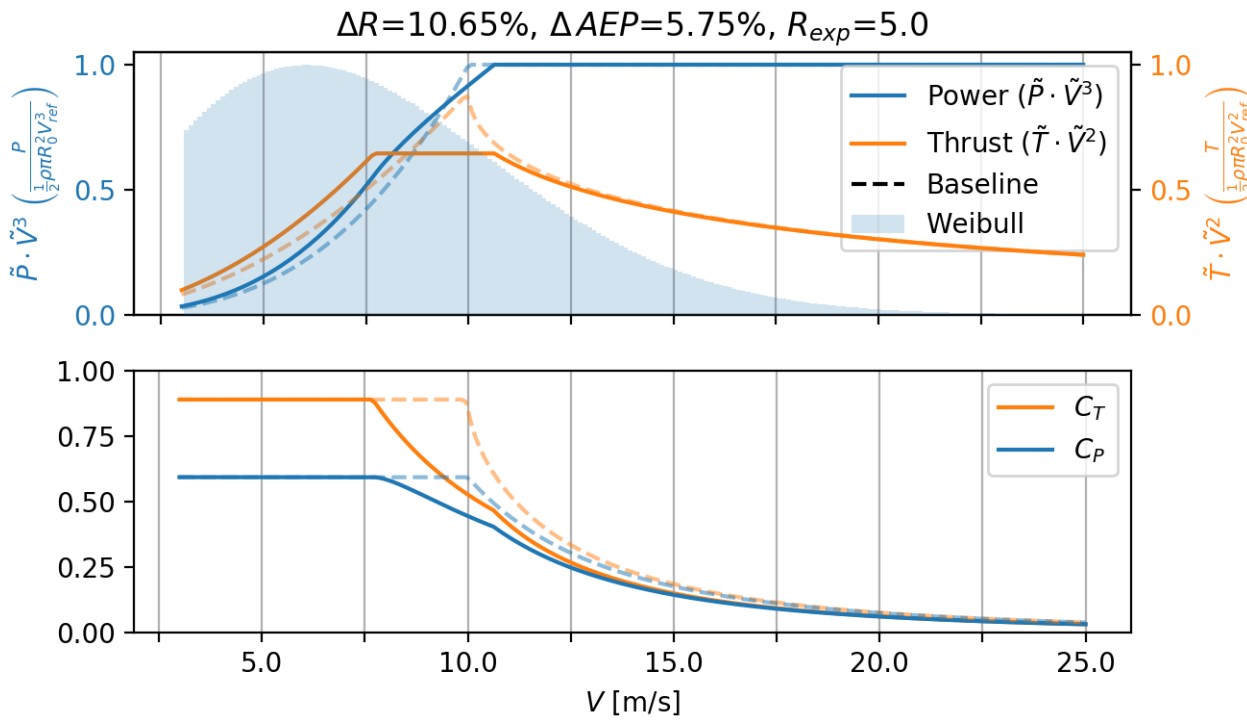

**Figure 11.** Power and thrust curve for AEP optimized rotor (solid lines) with DDLC exponent is $R_{exp} = 5$ which is equivalent to a constraint on $\delta_{tip}$. The dashed line is the baseline rotor optimized for max $C_P$ below rated power.

load for the optimized rotor for any $\tilde{V}$. The plot in figure 13 is similar to the plot in figure 5 with the difference being that it is the relative change in maximum loads, independently of wind speed at which it occurred. Comparing the two plots, one should note the range for the y-scale in the two plots, with figure 13 having the larger range. It also means that the relative change in the loads for the AEP-optimized rotor experiencing a larger relative change. But it also has the consequence that loads with $L_{exp} > R_{exp}$ grows faster especially for larger values of $R_{exp}$ ($> 5$). A summary for the $AEP$ optimized rotor can be seen in figure 14, where a table for 4 different design ($R_{exp} = 2, 3, 5, 6$) shows the relative change in $AEP$, radius, thrust, root-flap-moment and tip-deflection.

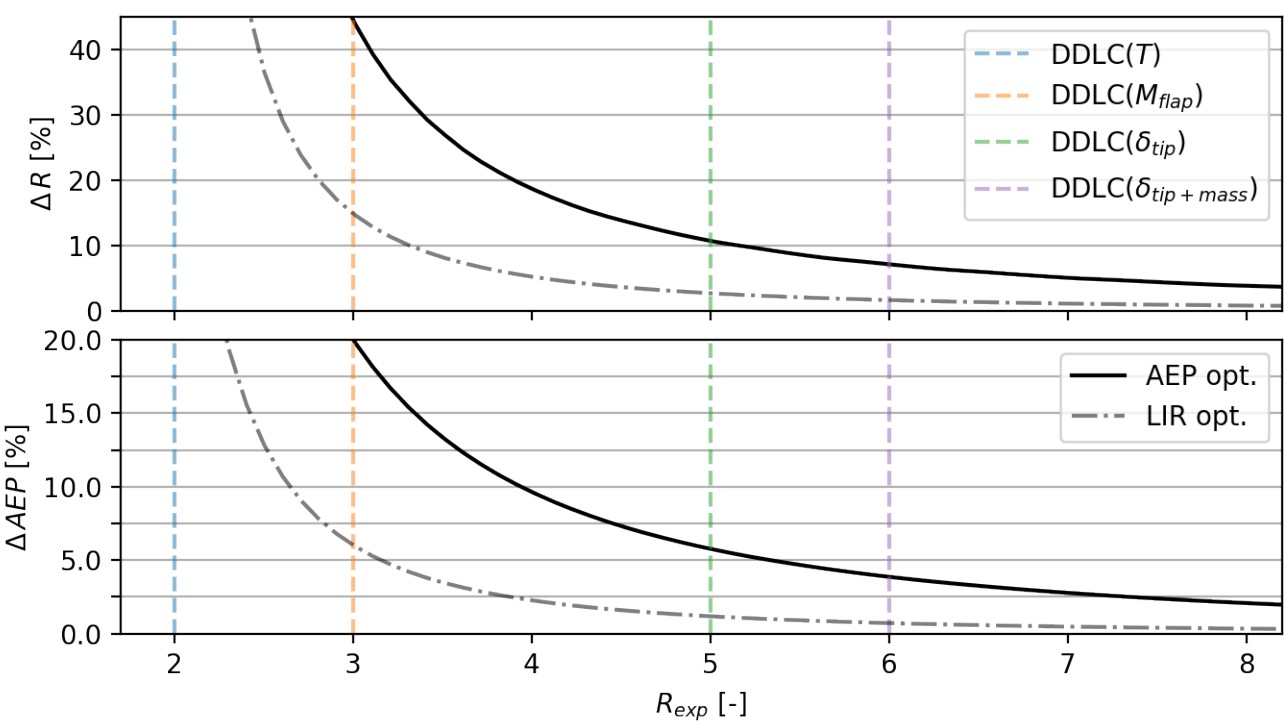

**Figure 12.** DDLC exponent ($R_{exp}$) vs. relative change in radius (upper graph, $\Delta R$) and relative change in AEP (lower graph, $\Delta A\tilde{E}P$). The plot both contains the changes for the case for Low-Induction-Rotor (LIR opt., black dashed-dot) and the AEP optimized (AEP opt., black solid). The changes in both $AEP$ and radius is much larger for the AEP optimized rotor.

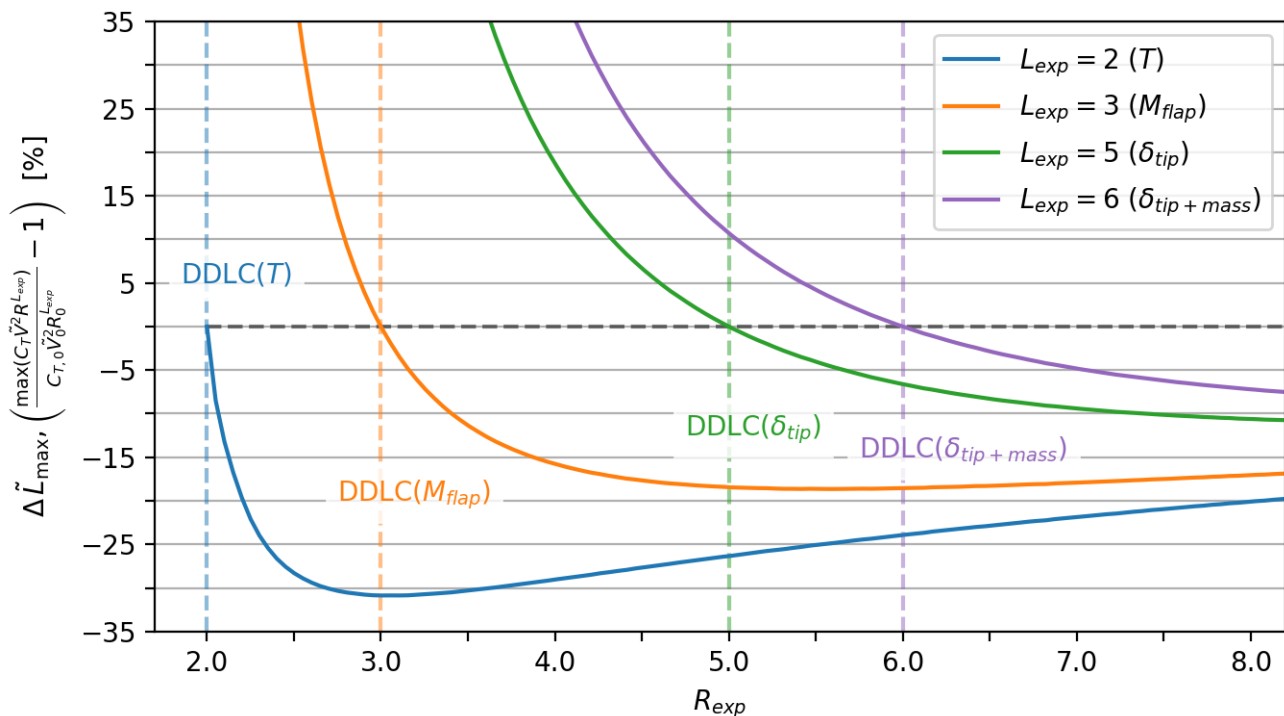

**Figure 13.** DDLC $R$-exponent ($R_{exp}$) vs. relative maximum load ($\Delta \tilde{L}_{\max}$). The plot looks similar to figure 5 but $\Delta \tilde{L}_{\max}$ is the change in maximum loading. As an example, when thrust ($T$) is $-30.8\%$ for $R_{exp} = 3$ it means that the maximum thrust (for any wind speed) is $30.8\%$ lower than the maximum thrust for the baseline (which happens just before rated wind speed). Notice that the range for the y-scale is much larger it this plot than for the power-capture optimized rotor. The potential reduction is more, but it comes with the consequence that $L_{exp} > R_{exp}$ grows faster even for high values of $R_{exp}$.

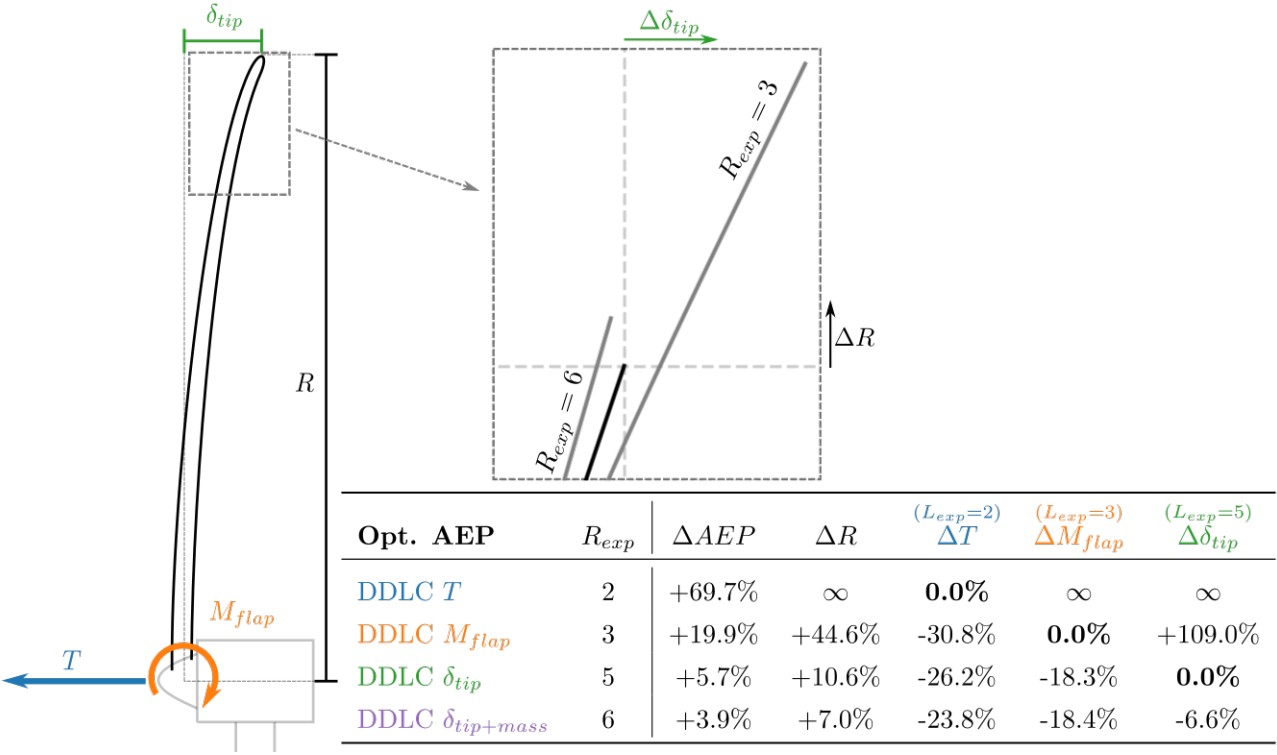

| Opt. AEP | $R_{exp}$ | $\Delta AEP$ | $\Delta R$ | $(L_{exp}=2)$ $\Delta T$ | $(L_{exp}=3)$ $\Delta M_{flap}$ | $(L_{exp}=5)$ $\Delta \delta_{tip}$ |
|---|---|---|---|---|---|---|
| DDLC $T$ | 2 | +69.7% | $\infty$ | **0.0%** | $\infty$ | $\infty$ |
| DDLC $M_{flap}$ | 3 | +19.9% | +44.6% | -30.8% | **0.0%** | +109.0% |
| DDLC $\delta_{tip}$ | 5 | +5.7% | +10.6% | -26.2% | -18.3% | **0.0%** |
| DDLC $\delta_{tip+mass}$ | 6 | +3.9% | +7.0% | -23.8% | -18.4% | -6.6% |

**Figure 14.** Sketch of a turbine with the load/structural response outlined. The zoomed figure shows the radius increase ($\Delta R$) and the change in tip-deflection ($\Delta \delta_{tip}$) for two different DDLCs (bold black line is the baseline). The table shows the relative change in power, radius and load/structural response for different DDLCs. $R_{exp} = 2$ is a thrust constraint design, $R_{exp} = 3$ is a flap moment constraint design, $R_{exp} = 5$ is a tip-deflection constraint design and $R_{exp} = 6$ is tip-deflection+constant mass constraint design.

## 4.4 Summary of Findings

In table 1 the tables shown in the figures 6,9 and 14 is summarized. It compares the different optimization's to each other.

| **Opt. PC** | $R_{exp}$ | $\Delta P$ | $\Delta R$ | $(L_{exp}=2)$ $\Delta T$ | $(L_{exp}=3)$ $\Delta M_{flap}$ | $(L_{exp}=5)$ $\Delta\delta_{tip}$ |
|---|---|---|---|---|---|---|
| DDLC $T$ | 2 | +50.0% | $\infty$ | **0.0%** | $\infty$ | $\infty$ |
| DDLC $M_{flap}$ | 3 | +7.6% | +11.6% | -10.4% | **0.0%** | +24.5% |
| DDLC $\delta_{tip}$ | 5 | +1.9% | +2.3% | -6.6% | -4.4% | **0.0%** |
| DDLC $\delta_{tip+mass}$ | 6 | +1.2% | +1.4% | -5.5% | -4.2% | -1.4% |

| **Opt. LIR** | $R_{exp}$ | $\Delta AEP$ | $\Delta R$ | $(L_{exp}=2)$ $\Delta T$ | $(L_{exp}=3)$ $\Delta M_{flap}$ | $(L_{exp}=5)$ $\Delta\delta_{tip}$ |
|---|---|---|---|---|---|---|
| DDLC $T$ | 2 | +49.7% | $\infty$ | **0.0%** | $\infty$ | $\infty$ |
| DDLC $M_{flap}$ | 3 | +6.0% | +14.9% | -12.9% | **0.0%** | +31.9% |
| DDLC $\delta_{tip}$ | 5 | +1.2% | +2.6% | -7.5% | -5.1% | **0.0%** |
| DDLC $\delta_{tip+mass}$ | 6 | +0.7% | +1.6% | -6.2% | -4.7% | -1.6% |

| **Opt. AEP** | $R_{exp}$ | $\Delta AEP$ | $\Delta R$ | $(L_{exp}=2)$ $\Delta T$ | $(L_{exp}=3)$ $\Delta M_{flap}$ | $(L_{exp}=5)$ $\Delta\delta_{tip}$ |
|---|---|---|---|---|---|---|
| DDLC $T$ | 2 | +69.7% | $\infty$ | **0.0%** | $\infty$ | $\infty$ |
| DDLC $M_{flap}$ | 3 | +19.9% | +44.6% | -30.8% | **0.0%** | +109.0% |
| DDLC $\delta_{tip}$ | 5 | +5.7% | +10.6% | -26.2% | -18.3% | **0.0%** |
| DDLC $\delta_{tip+mass}$ | 6 | +3.9% | +7.0% | -23.8% | -18.4% | -6.6% |

**Table 1.** Overview of the optimization results from optimizing Power-Capture (Opt. PC), Low-Induction-Rotor (Opt. LIR) and Annual Energy Production (Opt. AEP)

As seen from the tables the largest increase in $\Delta P/AEP$ is found using AEP-optimization, which also leads to the largest increase in rotor radius ($\Delta R$). It also shows that using thrust-clipping seems to be a better operational strategy than low-induction, as the design driving constraint can be met over a larger range of wind speeds and low-induction is only needed around maximum thrust and not at low wind speeds.

In all three optimization cases, the optimization of the design with thrust constraint (DDLC. T; $R_{exp} = 2$) leads to divergent values for $\Delta R$ and the loads. In all cases the result are found though investigating the limit value behavior when $R_{exp} \rightarrow 2$. Since this is not thought to be of much practical value, the details are not provided here.

## 4.5 Limitation of the study and possible improvements

The study shows that for a rotor constraint by a static aerodynamic DDL there is a benefit in lowering the loading and increasing the rotor size in terms of power/AEP. But as it was found by Bottasso et al. (2015) having a rotor with the same load constraint and increasing the radius does not mean that the cost is the same or that it is cost-optimal. They found that the increase in AEP did not compensate for the added cost by increasing the rotor radius. This problem of the cost-benefit is not directly addressed in this paper but by the DDLC $\delta_{tip+mass}$ a constraint where the mass is kept constant. It is thought to be a better approximation for a rotor with a fixed price - but this assumption needs to be tested.

Another issue that is not taken into account in this study is the influence of the turbines "self-weight". As was found by Sieros et al. (2012) the self-weight becomes more important for larger rotors. To accommodate for the added mass, a penalty could be added which should scale as $\tilde{R}$ or $\tilde{R}^3$ for "top head mass" and "static blade mass moment" respectively. As discussed above there could also be implemented a constraint that will keep the mass or the mass moment. Again this is a limitation of the study.

The fidelity of the models is also a limitation. Even though 1D-aerodynamic-momentum theory is a common approximation to do for first-order studies in rotor design it is well known that the constantly loaded rotor is not possible to realize and when losses are included the constantly loaded rotor is not the optimal solution anymore. At the same time if it was possible to decrease the load at the tip more than at the root it would lead to less tip-deflection than a constantly loaded rotor with a similar $C_T$. Extending the model to be able to handle radial load distribution is one way of detailing the model that could lead to even larger improvements. It could be done through the use of Blade Element Momentum (BEM) theory.

For modern turbine design, it is often the case that the structural design is determined by the aero-elastic extreme loads, such as extreme turbulence or gusts. With the simplicity of the models in this study, this is not taken into consideration. But if the extreme load happens in normal operation there will likely be a direct relationship between the steady- and extreme loads, meaning that a decrease in steady loads will also lead to a decrease in the extreme load. This is an assumption that should be tested in future work. If the design driving load is happening in non-operational conditions, e.g. extreme wind in parked conditions, grid loss, sub-component failure, etc. the analysis tool cannot be directly applied.

## 5 Conclusions

A first-order model framework for analysis of wind turbine rotors was developed based on aerodynamic 1D-momentum theory and Euler-Bernoulli-Beam theory. This framework introduces the concept of *Design Driving Load* (DDL) for which a generalized form has been developed where loads only differ by a scaling exponent $R_{exp}$, e.g. thrust scales as $R_{exp} = 2$, root-flap-moment as $R_{exp} = 3$ and tip-deflection as $R_{exp} = 5$. Despite the simplicity of the model, this study has shown important trends in how to design rotors for maximum power-capture. It has been shown that the potential increase in power-capture is very dependent on the relevant constraint, e.g. whether thrust is the constraining load or the more restrictive tip-deflection. Furthermore, it was concluded that the best way to design a rotor for increased power-capture using aero-elastic considerations is not to maximize $C_P$, but rather to relax $C_P$ and operate at lower loading (lower $C_T$). How much one should relax $C_P$ depends

on the chosen design driving constraint ($R_{exp}$). The results for optimizing for power-capture are summarized in Table 1 (Opt. PC).

The optimization of power-capture determines the best possible design for a given wind speed. By considering the *Annual Energy Production* ($AEP$), an optimal design across the range of operational wind speeds can be found for a given wind speed frequency distribution. Optimal $AEP$ was considered with two different approaches, namely *Low-Induction-Rotor* (LIR) and full $AEP$-optimization. For LIR, the $C_T$ value below rated power was set to the value found from power-capture optimization for the chosen $R_{exp}$. Then the radius was increased compared to the power-capture optimized rotor since it will reach rated power earlier with the same rotor size. A summary of the results can be seen in Table 1 (Opt. LIR).

For the full $AEP$-optimization, $C_T$ was allowed to take on any positive value below the Betz limit ($0 \leq C_T \leq 8/9$) for all wind speeds. The optimal $AEP$ is obtained for a rotor that operates in three distinct operational regimes:

- Operation with maximum power coefficient ($\max C_P$)

- Operation at constraint limit (constant thrust $T$)

- Operation at rated power

The results from the optimization are summarized in Table 1 (Opt. AEP). It shows significantly larger relative improvements in power/energy compared to power-capture and LIR optimized rotors. This comes at the cost of a larger increase in rotor radius. In the range where the optimum turbine operates at the constraint limit, the thrust curve is clipped (also known as peak shaving or force-capping). This is a control feature used for many contemporary turbines, so it is interesting that this study, independently of this knowledge, shows that thrust-clipping is a very efficient way to increase energy capture while observing certain load constraints. It is also the main reason behind the relatively large possible improvements in $AEP$, as the constraint limit is met over a larger range of wind speeds.

In spite of relatively crude model assumptions made, this paper provides profound insight into the trends of rotor design for maximum power/energy, e.g. the use of thrust clipping. As wind turbine rotors continue to develop towards larger diameters with slender (more flexible) blades, the type of design driving load constraints also evolves. With the present model framework, the conceptual implications of this development become clearer where an increase in AEP of up to 5.7% is possible compared to a traditional $C_P$ optimized rotor - without changing technology, using bend-twist coupling or other advanced features. Finally, this work has demonstrated an approach to formulate an optimization objective that couples power and load/structural response though the power-capture optimization. This approach may be extended into less crude model frameworks, e.g. by introducing radial variations in rotor loading.

*Author contributions.* KL came up with the concept and main idea, as well as made the analysis. All authors have interpreted the results and made suggestions for improvements. Also, some modeling has changed based on discussions between the authors. KL prepared the paper with revisions of all co-authors.

*Competing interests.* The authors declare that they have no conflict of interest.

*Acknowledgements.* We would like to thank Innovation Fund Denmark for funding the industrial PhD project, which this article is a part of. We would like to thank all employees at Suzlon Blade Sciences Center for being a great source of motivation with their interest in the results. We would like to thank all people at DTU Risø who came with valuable inputs.

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
