# Peer review of "Optimal relationship between power and design driving loads for wind turbine rotors using 1D models"

_Wind Energy Science, 2019_

## Referee Comment (RC1) · Anonymous Referee #1 · 21 Sep 2019

In general, the authors present an interesting topic with relevant results to the wind energy research community, and I enjoyed reading it. For instance, I like the theory part where the classical momentum theory expressions are used to find the optimal relationship between the aerodynamic power, thrust loading, and size of a wind turbine rotor. However, I have some minor comments that, in my opinion, might lead to an improvement of the overall paper quality. Therefore, I believe the paper merits publication in Wind Energy Science journal.
* * *
Minor Comments:

[Figure]

1.) There are several grammar errors here and there. I suggest a second reading using good grammar corrector.

2.) The introduction needs some revision to include more related works.

3.) The authors assumed that the change in CT does not lead to a proportional change in CP. Can the authors elaborate more on this assumption.

4.) The self-weight of the turbine is not taken into account in this study, the authors need to make this point clear in the manuscript including its impact on the general assumption used in the theory sections.

5.) The 1D-aerodynamic-momentum theory is considered as a first-order theory, the authors need to discuss broadly the benefit/shortcoming of using this theory instead for example using the Blade Element Momentum theory in the rotor design.

---

## Referee Comment (RC2) · Anonymous Referee #2 · 10 Oct 2019

**General comment**:

The paper is the fruit of a work based on an intense analytical derivation of the relationships, written in terms of some relevant turbine quantities (e.g thrust, flapwise moment, tip displacement), between a baseline rotor and one designed with the Low-Induction (LIR) concept .

In general, I believe to have inferred the idea underling the work, and I may imagine that the approach may have a potential utility in the context of LIR redesign, but honestly I have to say that the innovative content of the paper was not clearly stated in the

introduction nor in any other part of the paper. In fact:

- Line 44: "... it should be understood that the result presented here is not intended to be used directly for rotor design but to show a possible way to include structural/load constraints into the design process.". This concept is to be better explained, as it seems to refer to the scope of the paper and to hence the core of the research. Structural or load constraints are typically included into blade/rotor/turbine design processes, as witnessed by an extensive literature (see (Fleming et al., 2016; Perez-Moreno et al., 2016; Zahle et al., 2015) that the Authors already cited. To that list, I would personally add also the seminal work "Bottasso, Campagnolo, Croce, Multi-disciplinary constrained optimization of wind turbines, Multibody System Dynamics 27(1):21-53, 2012.".). From this point of view, given the current status of the introduction, the innovative content appears weak. Moreover, section 4.5 (I did appreciate that part with a thorough analysis on the limitations of the approach) lists a number of relevant issues of the developed methodology, which in my opinion should be previously introduced in the introduction. This could help readers understand the real innovative content of this paper.

- Line 50: "The constraints will not include the effect from aero-elastic extreme loads as it is thought to be out of scope for an analysis at this level. But it is expected that if the extreme loads happens in normal operation there should be a relationship between the steady and extreme loads.". The concept could be accepted, but often the ultimate loads come from extreme events such as gusts or during emergency shutdowns or even in parked conditions (where the treatment of this work is no more valid). This should be commented.

- Additionally, even assuming that the maximum loads for a rotor part comes from an operating condition, it is certainly possible that another subcomponent has an ultimate loads coming from a different condition (e.g. Extreme Wind in parked
condition). How can the proposed method handle this situation, given also the fact that different optimized rotors are associated to different loads (performance) through $R_{exp}$? This may appear as a strong limitiation of the applicability of the proposed method.

- If my comprehension of the work is correct, I would suggest to stress the fact that the methodology can be employed in a very preliminary stage of the design (or redesign) process, when simplified methodologies are always needed to select or define some global parameters (e.g. rotor radius). Additionally, the analysis is useful to comprehend the trend of thrust/loads/displacement as function of rotor radius, as already mentioned by Authors in the Conclusion. All in all, the innovative content is to be better clarify.

Finally, although they do not represent a "show-stop", there are many typos or sentences with grammatical errors, which should be corrected in a revised version of the paper, which is worth publishing. For example:

- Line 148: "... stiffness of the blade AT location ..."

- Line 149: "... the stiffness decreseS towards the tip ..."

- Line 278: "... Will not give the a design ..."

Therefore, I recommend to publish the paper only if Authors will accomodate the previous comments in a revised version of the paper. I also suggest some minor modifications, which may improve the manuscript.

**Minor comments**:

[Figure]

- Equation 4: This is just a detail: calling $C_P$ as "efficiency" is not correct from a theoretical standpoint as in the power equation one uses the indisturbed wind velocity in front of the rotor and the rotor area, which can be viewed as the area of the streamtube at turbine location. What I would like to say is that $1/2\rho V^3 \pi R^2$ does not represent the kinetic power ideally present in the wind because velocity $V$ and area $\pi R^2$ refer to different locations within the streamtube, hence $C_P$ is cannot be defined as "efficiency". In equation 4, this can be accepted as it is used in a generic way, but I suggest to modify line 83 at page 4 from "Efficiency is how much of the potential power the rotor can extract from the kinetic power of the wind." to "Efficiency is the part of the equation related to the power coefficient, representing the capability of the rotor to extract power from the wind." or something similar.

- Equation 11: $L_{exp}$ appears here for the first time but lacks of definition.

- Line 150: It should be appropriate to notices that a blade stiffness linearly proportional to the chord could be a strong approximation as the internal structure of a modern blade can be complex and could be even carachterized by discontinuities.

- Line 167: I was wondering whether this assumption be really necessary. In fact, one should be interesting only in having the same (or similar) tip displacement rather than the same deformation shape of the entire blade.

- Figure 7 and 8: It should be mentioned that the dashed lines refers to the baseline rotor and the solid ones to the LIR rotor.

- Figure 9: The symbol appearing in cells associated to $R_{exp} = 2$ and $\Delta R$, $\Delta M_{flap}$ and $\Delta\delta_{tip}$ is not clear.

- Figure 9: In the caption: Please, consider to add also the constraint of the design for $R_{exp}$ equal to 3 and 6, so as to provide a self-explaing figure.

- Line 323: "But for $\Delta AEP$ it will go towards a finite value", this is not clear looking at the plot. Please, explain.

- Caption of Fig. 13: "It is a similar plot to figure 5 but here it is for the AEP-optimized rotor and it is the change in the max load.". The sentence is not clear. Please, rephrase.

- Table 1: It is not straightforward to understand why for many conditions the "$\Delta$"-quantities go to infinity. I may suggest to add an explanation. Moreover, section 4.4 contains only the table and just a sentence. Consider the possibility to insert that content in a previous or subsequent section, or to extend the text with some comments.

- Line 388: "In spite of relatively . . . thrust clipping": the concept express in this sentence may be anticipated in the introduction within the context of the innovative content of work.

- Line 394: I agree with the possible inclusion of the radial variation of rotor loading, but what about a the use of a more realistic relationship between $C_P$ and $C_T$? In fact a wind turbine may operate close to $C_T = 8/9$ but far from the Betz optimal $C_P$.

---

## Author Comment (AC1) · 14 Nov 2019

Thank you for taking the time to read our paper. Your comments are appreciated and we believe that they have made the manuscript better.

The following is the author's answer to the minor comments. The *italic* text is the referee question/comment the following text is the author's answer/comment. The **(bold)** is the page - (p. #) and line number (l. #) in the document: DIFF_Optimal_power_capture_for_wind_turbines_with_design_driving_loads.pdf attached to this comment where the change has been highlighted.

*1.) There are several grammar errors here and there. I suggest a second reading using good grammar corrector.*
We have been though the paper a couple of times and the grammar should be better now.

*2.) The introduction needs some revision to include more related works.*
It is not clear to the authors if the referee has a specific part of the literature that he/she thought was missing?! As pointed out by another referee the work by Bottasso et al., 2010 (Multi-disciplinary constrained optimization of wind turbines) is a seminal work when talking about MDAO for wind turbine design. It is therefore added to the list of MDAO references. (**p. 2, l. 30**)
We also added the work by Buck and Garvey (2015a) discussing "thrust clipping" to the introduction. Their work was mentioned later in the paper, but it was thought that an earlier introduction would be better. (**p. 2, l. 50**)

*3.) The authors assumed that the change in CT does not lead to a proportional change in CP. Can the authors elaborate more on this assumption.*
It is an assumption that is a direct consequence of using 1D-momentum theory. It is best seen in equation (3) **(p. 4, l. 106)** where the classical equations for $C_T = 4a(1-a)$ and $C_P = 4a(1-a)^2$ is combined to an expression for the relationship between $C_P$ as a function of $C_T$.

*4.) The self-weight of the turbine is not taken into account in this study, the authors need to make this point clear in the manuscript including its impact on the general assumption used in the theory sections.*
Indeed, the self-weight is not part of the optimization presented in this paper. This was mentioned in section 4.5 (Limitation of the study and possible improvements) **(p. 28, l. 415-419)**. But this is at the end of the article and as also pointed out by others the

limitations of the study should have been mentioned in the introduction to make it clear for the reader which level of detail the study deals with. To accommodate this we have added a further discussion about the limitation of the study to the introduction, where the self-weight is also mentioned. **(p. 3, l. 82-87)**

*5.) The 1D-aerodynamic-momentum theory is considered as a first-order theory, the authors need to discuss broadly the benefit/shortcoming of using this theory instead for example using the Blade Element Momentum theory in the rotor design.*
Related to the previous comment, we have now added a further discussion about the limitations of the study to the introduction. **(p. 3, l. 66-87)** This should clarify the intent of the paper as a tool for rotor analysis in the initial stage.
Furthermore, Blade Element Momentum theory is thought to be an extension of the 1D-aerodynamic-momentum theory where losses are taken into account and the load can be varied radially as discussed in section 4.5 **(p. 28, l. 420-426)**. The authors are currently working on generalizing the method for radial load variations.

Please also note the supplement to this comment:
https://www.wind-energ-sci-discuss.net/wes-2019-28/wes-2019-28-AC1-supplement.zip

---

## Author Comment (AC2) · 14 Nov 2019

Thank you for taking the time to read our paper. Your comments are appreciated and we believe that they have made the manuscript better.

The following is the author's answer to the referee's questions/comments. The *italic* text is the referee question/comment the following text is the author's answer/comment. The **(bold)** is the page - (p. #) and line number (l. #) in the document : DIFF_Optimal_power_capture_for_wind_turbines_with_design_driving_loads.pdf attached to this comment where the change has been highlighted.

**General Comment**

*The innovative content of the paper was not clearly stated*

After rereading the introduction with this in mind the authors agree with the referee - the innovative content is not clearly stated. To make this clear we added a paragraph about where these results can be applied **(p. 2-3, l. 54-65)**. The sentence which the referee mention ("... it should be understood that the result presented here is not intended to be used directly for rotor design ..") is also taken out, as the authors meant detailed design like blade plan-form but this was not clear from the text.

*Limitations in introduction*

From the comment: *... section 4.5 ... which in my opinion should be previously introduced in the introduction. This could help readers understand the real innovative content of this paper.*

The authors agree with this point and it has been accommodated by the added discussion in the introduction **(p. 3, l. 66-87)**

*Additional MDAO reference*

As mentioned by the referee the work by "Bottasso, Campagnolo, Croce, Multi-disciplinary constrained optimization of wind turbines, Multibody System Dynamics" is a seminal work for the use of MDAO to wind turbines and it should be part of the list of references and it is therefore added to the list of references. **(p. 2, l. 30)**

*Aero-elastic extreme loads*

As mentioned earlier, we have added further discussion of the limitation of the study. Here we also discuss the limitation of aero-elastic extreme loads. **(p. 3, l. 66-73)**

**Minor comments**

*Calling $C_P$ as "efficiency" is not correct from a theoretical stand point*
The authors did not consider this fact before it was pointed out by the referee. The suggested change has been adapted **(p. 4, l. 117)**.

*Equation 11: $Lexp$ appears here for the first time but lacks of definition.*
Indeed, both $L_{exp}$ as well as $\tilde{L}$ has not been defined at this point. It is written here for later reference, which has been written in the subsequent text **(p. 7, l. 155)**.

*Line 150: It should be appropriate to notices that a blade stiffness linearly proportional to the chord could be a strong approximation as the internal structure of a modern blade can be complex and could be even characterized by discontinuities.*
The assumption of $EI \propto c$ is a crude approximation considering the complex structure that is a wind turbine blade. With that said, the model with $EI \propto 1/r$ is found to match fairly well with modern wind turbine blades capturing the behavior that $EI$ becomes smaller for larger $r$. It is thought that the fidelity of this approximation is at least on the same order as the other models used in this study.

*Line 167: I was wondering whether this assumption be really necessary. In fact, one should be interesting only in having the same (or similar) tip displacement rather than the same deformation shape of the entire blade.*
It is an interesting point, and the authors did not think of this case. As the referee points out, assuming that $\delta_{shape}$ is the same when increasing $R$ is a sufficient assumption - but it is not necessary. The more general assumption is now added to the paper **(p. 9, l. 202)**.

*Figure 7 and 8: It should be mentioned that the dashed lines refers to the baseline rotor and the solid ones to the LIR rotor.*

[Figure]

A comment about the dashed lines is now added to all the power-curves containing a baseline curve. (figure 7, 8, 10 and 11)

*Figure 9: The symbol appearing in cells associated to $R_{exp} = 2$ and $\Delta R$, $\Delta M_{flap}$ and $\Delta \delta_{tip}$ is not clear.*
The authors agree that the $\infty$ symbol in figure 6,9 and 14 were not clear with a smaller font than the numbers. The figures have been updated together with tables in table 1, to make the content of the tables clearer.

*Figure 9: In the caption: Please, consider to add also the constraint of the design for $R_{exp}$ equal to 3 and 6, so as to provide a self-explaining figure.*
We agree with the referee that this would make the figure easier to understand, and it was added to the caption.

*Line 323: "But for $\Delta AEP$ it will go towards a finite value", this is not clear looking at the plot. Please, explain.*
&
*It is not straightforward to understand why for many conditions the "$\Delta$"-quantities go to infinity. I may suggest to add an explanation.*
There is indeed no explanation for the limiting cases when $R_{exp} \rightarrow 2$. The authors agree that it was confusing not to mention why this is the case. We have added a comment that the result is found by investigating the limit $R_{exp} \rightarrow 2$. **(p. 14, l. 281)(p. 17, l. 346)(p. 22, l. 387)(p. 27, l. 405)**. The explanation for this case was thought to be "complicated" and that it would overshadow the results. Especially considering that this limit is not of much practical value. We have attached a separate pdf appendix to this comment ("Limit_Rexp->2.pdf") where the limit values are found for all the three optimization case. We do not plan to add this appendix to the article since it is thought to complicate the understanding of the paper.

*Caption of Fig. 13: "It is a similar plot to figure 5 but here it is for the AEP-optimized rotor and it is the change in the max load.". The sentence is not clear.Please, rephrase.*
After rereading the caption the authors agree and the sentence is hard to interpret. It was rephrased. **(p. 25, fig. 13)**

*Section 4.4 contains only the table and just a sentence. Consider the possibility to insert that content in a previous or subsequent section, or to extend the text with some comments.*
The authors added some comments to the section comparing the three optimization methods. **(p. 27, l. 401-407)**

*Line 388: "In spite of relatively ... thrust clipping": the concept express in this sentence may be anticipated in the introduction within the context of the innovative content of work.*
A paragraph mentioning the concept of thrust-clipping and the study by Buck and Garvey (2015a) was added to the introduction. **(p. 2, l. 47-53)**

*Line 394: I agree with the possible inclusion of the radial variation of rotor loading, but what about the use of a more realistic relationship between $C_P$ and $C_T$? In fact a wind turbine may operate close to $CT = 8/9$ but far from the Betz optimal $C_P$*
It is true that in practice a wind turbine will not reach close to Betz-limit. This is often a consequence of a non-constantly loaded rotor - which is an assumption in this study. For a non-constantly loaded rotor, the loads are not directly related thought $C_T$ and $R$ and the method in this study can not directly be applied. The authors are currently working on generalizing the framework for the radially varying case.

Please also note the supplement to this comment:
https://www.wind-energ-sci-discuss.net/wes-2019-28/wes-2019-28-AC2-
supplement.zip
* * *